# Room-temperature quantum emission from interface excitons in mixed-dimensional heterostructures

N. Fang [1] ✉, Y. R. Chang [1], S. Fujii [2,3], D. Yamashita [2,4], M. Maruyama [5], Y. Gao [5], C. F. Fong [1], D. Kozawa [1,2,6], K. Otsuka [1,7], K. Nagashio [8], S. Okada [5] & Y. K. Kato [1,2] ✉

The development of van der Waals heterostructures has introduced unconventional phenomena that emerge at atomically precise interfaces. For example, interlayer excitons in two-dimensional transition metal dichalcogenides show intriguing optical properties at low temperatures. Here we report on room-temperature observation of interface excitons in mixed-dimensional heterostructures consisting of two-dimensional tungsten diselenide and one-dimensional carbon nanotubes. Bright emission peaks originating from the interface are identified, spanning a broad energy range within the telecommunication wavelengths. The effect of band alignment is investigated by systematically varying the nanotube bandgap, and we assign the new peaks to interface excitons as they only appear in type-II heterostructures. Room-temperature localization of low-energy interface excitons is indicated by extended lifetimes as well as small excitation saturation powers, and photon correlation measurements confirm antibunching. With mixed-dimensional van der Waals heterostructures where band alignment can be engineered, new opportunities for quantum photonics are envisioned.

The discovery of van der Waals (vdW) materials, including two-dimensional (2D) transition metal dichalcogenides (TMDs) and graphene, has brought about a revolution in the assembly of artificial heterostructures by allowing for the combination of two different materials without the constraints of lattice matching. Such an unprecedented level of flexibility in heterostructure design has led to the emergence of novel properties not seen in individual materials. A prime example is twisted bilayer graphene at magic angles, which exhibits exotic phases such as correlated insulating states[1] and superconductivity[2]. Another notable development is the stacking of two TMDs, resulting in the observation of unique excitons known as interlayer excitons, characterized by electrons and holes located in

separate layers[3–6]. The spatially indirect nature of interlayer excitons imparts them with distinct properties, including long exciton lifetimes[3], extended diffusion lengths[7], large valley polarization[8], and significant modulation by moiré potentials[9,10].

The existing vdW heterostructures comprise 2D materials with similar lattice structure, excitonic characteristics, and inherently identical dimensions. Development of vdW heterostructures that encompass lower dimensional materials may give rise to unique interface exciton states resulting from the mixed dimensionality. Carbon nanotubes (CNTs), a typical one-dimensional (1D) material, are ideal for such heterostructures as they have all bonds confined to the tube itself[11,12]. CNTs interact with 2D materials through weak vdW

[1]Nanoscale Quantum Photonics Laboratory, RIKEN Cluster for Pioneering Research, Saitama, Japan. [2]Quantum Optoelectronics Research Team, RIKEN Center for Advanced Photonics, Saitama, Japan. [3]Department of Physics, Keio University, Yokohama, Japan. [4]Platform Photonics Research Center, National Institute of Advanced Industrial Science and Technology (AIST), Ibaraki, Japan. [5]Department of Physics, University of Tsukuba, Ibaraki, Japan. [6]Research Center for Materials Nanoarchitectonics, National Institute for Materials Science, Ibaraki, Japan. [7]Department of Mechanical Engineering, The University of Tokyo, Tokyo, Japan. [8]Department of Materials Engineering, The University of Tokyo, Tokyo, Japan. ✉e-mail: nan.fang@riken.jp; yuichiro.kato@riken.jp

forces, resulting in well-defined, atomically sharp interfaces[13,14]. The chirality-dependent bandgap of CNTs can be utilized to tune the band alignment as demonstrated in exciton transfer process[15], allowing for unambiguous identification of excitonic states at the 1D-2D interface.

Here we report on the observation of multiple emergent excitonic peaks in the 1D-2D CNT/tungsten diselenide (WSe$_2$) heterostructures at room temperature. These peaks appear exclusively at the interface region with a broad energy range lower than CNT E$_{11}$ states, and their dependence on the chirality of CNTs and the layer number of WSe$_2$ is investigated. The emergence of the peaks is found to be highly correlated with the band alignment, and they are interpreted as interface excitons. Prominent linear polarization, low excitation saturation power, and a long lifetime are characteristic of low-energy interface excitons, suggesting strong confinement. Through photon correlation measurements, room-temperature antibunching has been confirmed. These findings expand the existing concept of spatially indirect excitons based on 2D heterostructures to 1D systems, demonstrating significant potential of the interface excitons for nanophotonics and quantum information processing.

## Results and discussion

### Emerging peaks in 1D-2D mixed-dimensional heterostructures

The CNT/WSe$_2$ heterostructures under investigation are entirely freestanding to preclude substrate effects[5,16], as depicted in Fig. 1a, b. High-quality CNTs are initially grown over trenches (see Methods and Supplementary Fig. 1), followed by placement of a WSe$_2$ flake upon the tubes using the anthracene-assisted transfer technique[17,18]. We perform photoluminescence excitation (PLE) measurements to determine the

chirality before transfer for all samples. A (9,4) CNT is selected as a representative case, forming type-II band alignment with WSe$_2$[15]. Such alignment should establish emergent excitonic states between the CNT conduction band minimum and the WSe$_2$ valence band maximum (Fig. 1c). The formation of the indicated indirect excitons generally requires charge transfer, which is plausible as exciton transfer has been observed in similar heterostructures with type-I band alignment[15].

Room-temperature photoluminescence (PL) spectroscopy is employed to investigate the excitonic states present within the heterostructure[19–21]. The PL spectrum of the pristine, suspended (9,4) CNT displays a singular peak at 1.143 eV, corresponding to the E$_{11}$ transition (Fig. 1d)[11]. After the transfer of the monolayer WSe$_2$ flake, the CNT E$_{11}$ peak is redshifted to 1.102 eV (Fig. 1e) as a result of the dielectric screening effect[14], indicating intimate contact between the two materials. Notably, two excitonic peaks arise with energies lower than E$_{11}$. These peaks cannot be attributed to the suspended monolayer (1L) WSe$_2$ emission as only the A exciton peak at 1.658 eV is expected (see Supplementary Fig. 2). The transfer process is not expected to introduce defects, since CNT/hexagonal boron nitride heterostructures prepared in a similar manner does not exhibit such low-energy peaks[14], suggesting the existence of emergent excitonic states in the CNT/WSe$_2$ heterostructure. The newly emerged peaks are initially unstable and exhibit temporal blinking (inset in Fig. 1e). Such PL evolution is not observed in sp$^3$ defects, which exhibit more stable emission once formed[22,23]. Following a spectral development involving fluctuations of the peaks, the unstable excitonic states vanish and stable states remain for which we perform the subsequent measurements (see Supplementary Fig. 3). Figure 1f presents the stable PL

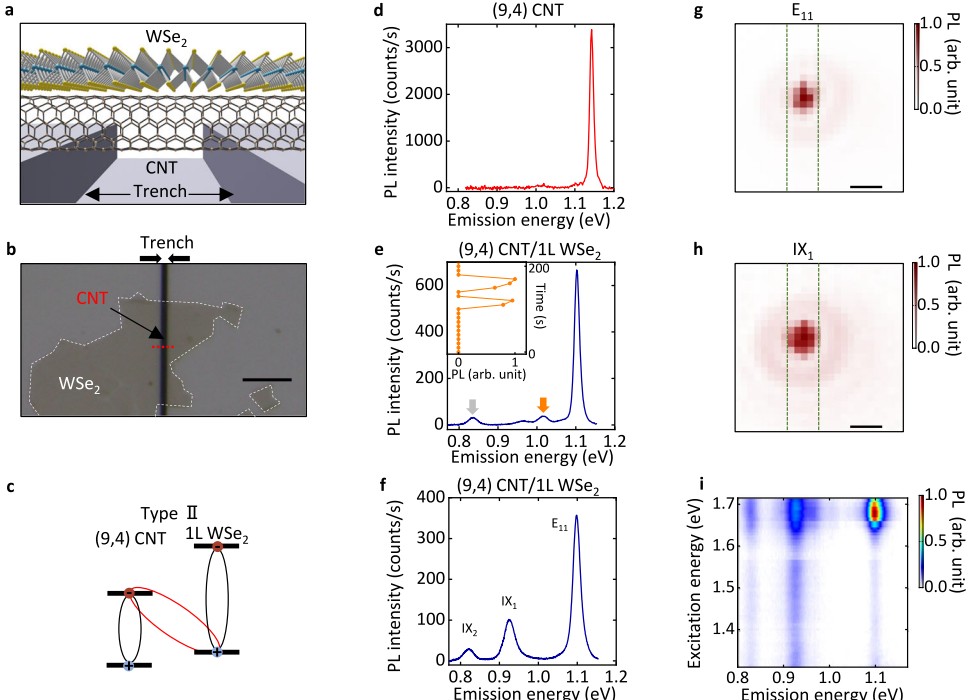

**Fig. 1 | Emergence of interface excitons in 1D-2D heterostructures. a** A schematic of a suspended carbon nanotube (CNT)/WSe$_2$ heterostructure. **b** An optical microscope image of the (9,4) CNT/1L WSe$_2$ heterostructure. The substrate is SiO$_2$/Si. The red and white dashed lines indicate CNT and WSe$_2$, respectively. The scale bar represents 5 μm. **c** The band diagram of the (9,4) CNT/1L WSe$_2$ heterostructure, illustrating the excitons in CNT and WSe$_2$ as well as the interface exciton. **d** A photoluminescence (PL) spectrum of the pristine, suspended (9,4) CNT prior to WSe$_2$ transfer. **e** A PL spectrum immediately following the transfer of 1L WSe$_2$ to form the heterostructure. Emergent peaks are denoted by IX. Arrows indicate unstable IX peaks. The inset depicts the time-trace of the integrated PL intensity for

the IX peak indicated by the orange arrow. The peaks at 1.011 and 0.962 eV in **d**, **e** are K-momentum exciton peaks. **f**, A PL spectrum from the same sample after approximately eight hours of optical measurements. E$_{11}$ indicates the CNT E$_{11}$ state, and IX$_1$ and IX$_2$ indicate the emergent excitonic states. **g**, **h** PL intensity maps of E$_{11}$ **g** and IX$_1$ **h**. The edges of the trench are indicated by the green broken lines. The scale bar represents 1 μm. **i** A photoluminescence excitation (PLE) map of the (9,4) CNT/1L WSe$_2$ heterostructure. The excitation power is 4.5 and 5 μW for **f** and **i**, and 10 μW otherwise. The excitation energy is 1.730 eV for **d**, **e**, 1.680 eV for **f**, and 1.653 eV for **g**, **h**.

spectrum, featuring two prominent peaks at 0.924 eV and 0.821 eV, denoted as $IX_1$ and $IX_2$, respectively. With an energy difference of approximately 0.278 eV from $E_{11}$, $IX_2$ is at a lower energy than any reported dark or defect states in (9,4) CNTs[24–26]. Such low-energy peaks are also observed in other monolayer $WSe_2$ heterostructures with type-II band alignment (see Supplementary Fig. 4). We hypothesize these states to be interface excitons, which could possess substantially lower excitonic energies as determined by the heterostructure band alignment.

The spatial and spectral correlation between $E_{11}$ excitons and IXs is investigated by conducting PL imaging measurement and PLE spectroscopy. Figure 1g is an integrated PL image from $E_{11}$, exhibiting strong signal above the trench due to the quenching from the silicon dioxide ($SiO_2$)/silicon (Si) substrate. Both $IX_1$ and $IX_2$ peaks are observed precisely at the position of the $E_{11}$ peak, as demonstrated by the $IX_1$ and $IX_2$ images displayed in Fig. 1h and Supplementary Fig. 5, respectively. Such spatial overlap with the CNT emission indicates that IX peaks cannot be explained by emission from randomly distributed defect states within $WSe_2$. In the PLE map (Fig. 1i), $E_{11}$ shows a strong response to excitation energy, which is identified as CNT $E_{22}$ transition (Supplementary Fig. 6). Similar response as $E_{11}$ is observed for $IX_1$ and $IX_2$ peaks, implying that the carriers forming the IXs are supplied from the CNT. In comparison, we do not observe a clear signature of the $WSe_2$ A exciton peak in the PLE map. Considering IXs only emerge upon transfer of the $WSe_2$ flake, the spatial and spectral overlap with CNT supports the hypothesis that they originate from the interface.

## Band alignment effect on IX peaks

Since interface excitons form between the two materials, manipulating the heterostructure band alignment should affect the IXs[6,27]. It is possible to vary the CNT bandgap (Fig. 2a) by studying different CNT chiralities (Fig. 2b), whereas $WSe_2$ bandgap can be altered (Fig. 2c) through the layer number (Fig. 2d, Supplementary Fig. 2).

We first investigate the chirality dependence, which significantly modulates the CNT bandgap. The band alignment is systematically tuned by utilizing CNT/$WSe_2$ heterostructures with different CNT chiralities as illustrated in Fig. 2b. $WSe_2$ layer numbers ≤ 4 are used for heterostructures since IX peaks can be observed as shown in the case for (9,4) CNTs with bilayer (2L) and quadlayer (4L) $WSe_2$. It is noted that the valance band maximum of $WSe_2$ changes from the K point to the Γ point with increasing the layer number, but correlation with the behavior of IX peaks cannot be observed. Multiple IX peaks appear in (9,4), (12,1), (8,6), (8,7), and (14,0) CNTs, which have large $E_{11}$ energies and therefore large bandgaps. This is consistent with the expectation that a large bandgap is favorable for type-II band alignment as depicted in Fig. 2a. It is noted that $sp^3$ defects in CNTs generally introduce doublet peaks[23], different from the numerous IX peaks observed here (see Supplementary Fig. 3). The observed IX peaks span a broad energy range within the telecommunication wavelengths. Note that the low-energy peaks approach the edge of the spectral window, suggesting the possibility of lower-energy IX peaks existing beyond our current detection capability. Meanwhile, the highest energy peak in each chirality is located close to $E_{11}$, with a difference of ~ 0.05 eV. Remarkably, a further decrease in the bandgap leads to the disappearance of IXs (Supplementary Fig. 7). The presence of the IX peaks is determined by chirality, consistent with the transition in band alignment from type-II to type-I that is observed in a density functional theory simulation[15]. We note that for $sp^3$ defects, there exists no such transition and any chirality CNT can form $sp^3$ defects[25]. We therefore identify IXs as interface excitons.

The IX peaks in the PL spectra from various heterostructures are summarized in Fig. 2e by plotting the number of the peaks (N) observed during time-trace measurements as a function of $E_{11}$ (see Supplementary Table 1 for the list of samples). Two distinct regions can be seen below and above 0.94 eV, corresponding to type-I

and type-II alignment, respectively (see Supplementary Fig. 8). IX peaks are absent for type-I band alignment, whereas numerous peaks appear for type-II alignment. Exciton transfer observed in similar heterostructures show anticorrelation with the appearance of the IX peaks, consistent with the band alignment transition (Supplementary Fig. 9)[15].

The dependence of IXs on the number of $WSe_2$ layers (Fig. 2c) is more subtle, since the number of layers does not significantly modulate the bandgap in comparison to CNT chirality (Supplementary Fig. 10). For example, (9,4) CNT/$WSe_2$ heterostructures are consistent with complete type-II band alignment irrespective of the $WSe_2$ layer number (Figs. 1f and 2b). We therefore study (10,5) CNT/$WSe_2$ heterostructures located at the band alignment transition[15], as they should be sensitive to the small changes in $WSe_2$ bandgap. The PL spectrum of the 1L $WSe_2$ heterostructure is presented in Fig. 2d, which does not show any observable IX peaks. In contrast, the trilayer (3L) $WSe_2$ heterostructure reveals a discernible IX peak in between the $E_{11}$ exciton and the trion (T) peaks. Two IX peaks appear for the 4L $WSe_2$ heterostructure, with increased PL intensity for the higher energy IX peak and an additional lower energy IX peak besides the trion peak. This layer-number dependent behavior of the IX peaks can be explained by the band alignment transition, as shown in Fig. 2c.

## Optical properties of the interface excitons

The interface excitons display several distinct features different from the $E_{11}$ excitons. In Fig. 3a, we first examine the emission polarization dependence of $IX_1$ and $IX_2$ for the (9,4) CNT/1L $WSe_2$ sample used in Fig. 1. PL from IXs exhibits near ideal linear polarization of > 95%, consistent with confinement in the 1D channel[11,28]. We also note that the polarization angle of $IX_1$ and $IX_2$ deviates from that of $E_{11}$ polarization by 13.7°, potentially suggesting some distortion of the optical dipole moment in the IXs (see Supplementary Fig. 11). Owing to the use of a conventional normal-incidence photoluminescence setup which predominantly detects in-plane dipoles, the vertical component remains unresolved in our experiments.

Time-resolved PL is then performed for $E_{11}$ and $IX_2$ excitons, as the lifetime of IXs is expected to be long due to the spatially indirect nature[3,7]. PL decay curves corresponding to $E_{11}$ and $IX_2$ are measured as shown in Fig. 3b, and the lifetime is extracted by reconvolution fitting using the instrument response function (IRF). Two decay components are obtained for the $E_{11}$ PL decay curve, as is the case for suspended CNTs: A main fast component with a lifetime of 59 ps associated with the bright states, and a small slow component with a lifetime of 646 ps associated with the dark states[12]. In contrast, only one decay component is observed for $IX_2$ with a long lifetime of 673 ps, consistent with the reduced optical dipole moment.

The interface excitons exhibit considerably bright emission at low excitation powers as shown in Fig. 3c. At a low power of 0.04 μW, both $IX_1$ and $IX_2$ display bright emission with the $IX_1$ PL even exceeding the $E_{11}$ PL. Such high intensity emission from interface excitons is unexpected at room temperature. Generally, indirect excitons exhibit weak PL emission due to diminished dipole coupling, which is the case for 2D-2D heterostructures where interlayer excitons can hardly be observed at room temperature[3,4,6]. The evident PL from interface excitons in our system can be ascribed to two primary factors. Firstly, we employ a fully suspended structure, which reduces the substrate-induced screening effect and helps sustain the dipole strength. Secondly, the wavefunction of π-orbitals in CNTs, which extend significantly out of the tube, could reduce the spatially indirect nature of the interface excitons.

Emission intensity of the interface excitons exhibit intriguing power dependence as shown in Fig. 3d. Both $IX_1$ and $IX_2$ PL nearly saturate with a low threshold power of approximately 0.6 μW, while the $E_{11}$ PL increases substantially. The saturation observed is much more pronounced compared to interlayer excitons in 2D-2D

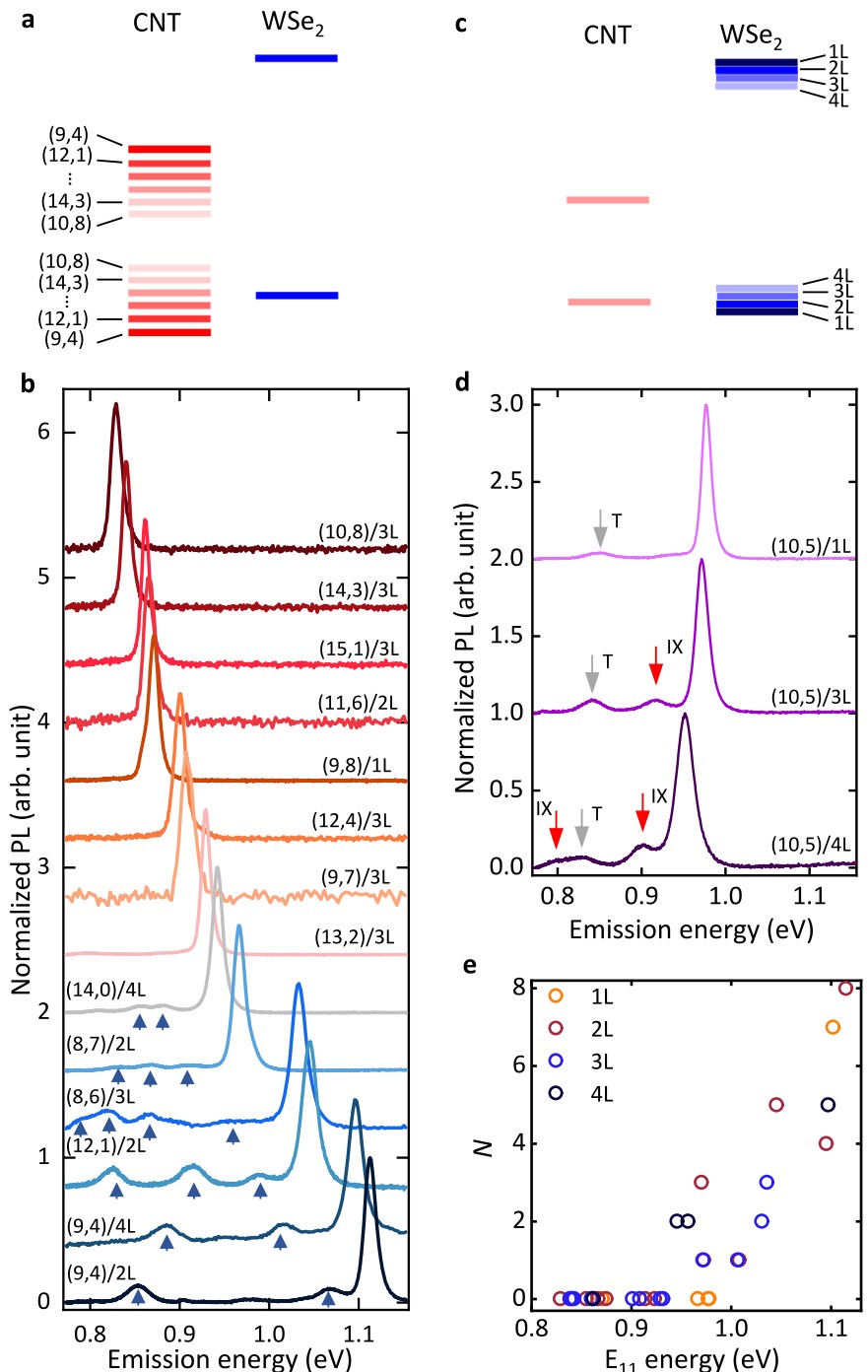

**Fig. 2 | Band alignment effect on IXs. a** Band diagram of heterostructures constructed from CNTs with various chiralities. **b** Chirality-dependent PL spectra of 1D-2D heterostructures. We define a heterostructure nomenclature where (9,4) CNT/2L WSe$_2$ is represented by (9,4)/2L, with other samples following the same convention. Excitation energy is adjusted to CNT E$_{22}$ state for each heterostructure. Excitation power values are 4, 6, 5, and 5 µW for (9,4)/2L, (12,1)/2L, (8,7)/2L, and (14,3)/3L heterostructures, respectively, and 10 µW for other samples. Arrows indicate IX peaks. Samples exhibiting IX peaks may also host other unstable IX peaks on a longer timescale, which is not indicated here. The peaks from K-momentum states are carefully checked and excluded in the analysis of IXs.

**c** Band diagram of heterostructures constructed from WSe$_2$ with different layer numbers. **d** PL spectra from (10,5)/1L, (10,5)/3L, and (10,5)/4L samples. The excitation energy is adjusted to 1.653 eV and the excitation power is 5 µW for the (10,5)/4L sample and 10 µW for others. Red arrows indicate IX peaks while the gray arrow denotes trion peaks T. **e** The plot of the number of the IX peaks $N$ as a function of E$_{11}$ energy for all samples. It should be noted that $N$ encompasses all IX peaks observed during measurements, and some of them are not illustrated in **b**. More spectra of IX peaks are shown in Supplementary Figs. 3, 7, and 12. The energies of all IX peaks can be found in Supplementary Fig. 8.

heterostructures[3]. It is suspected that the IXs are further confined to a lower dimension, that is, 0D. Interface excitons in CNTs may be more readily localized than interlayer excitons in 2D-2D heterostructures because of the lower dimensionality. In general, localized states show much stronger saturation behavior than free states because of the

state-filling effects[29,30]. In other samples where interface excitons also emerge, we find that the saturation behavior depends on their energies (see Supplementary Fig. 12). The IX peaks with energies substantially lower than the E$_{11}$ energy display stronger saturation behaviors. This could be explained by the deeper trap potential in the confinement of

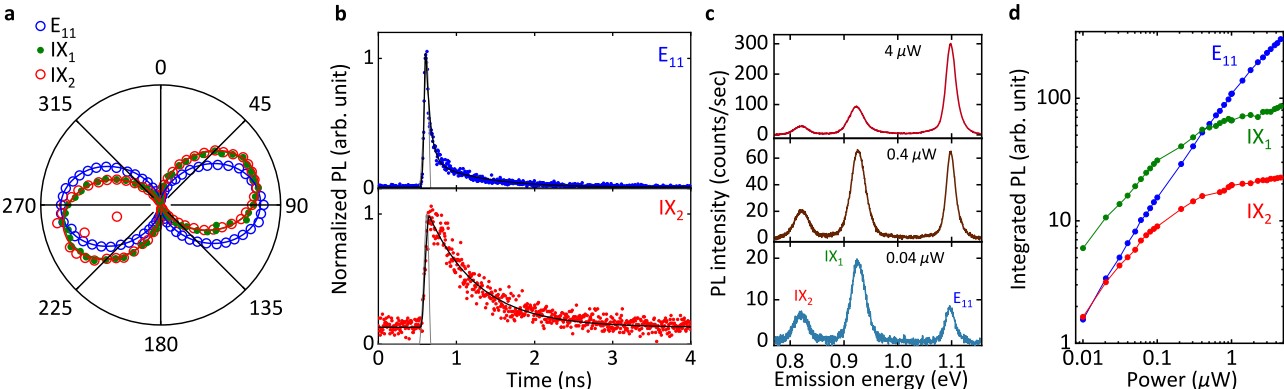

**Fig. 3 | Optical properties of low-energy interface excitons. a** Emission polarization dependence of PL emission from $E_{11}$ (blue circles), $IX_1$ (green dots), and $IX_2$ (red circles). PL emission is plotted as a function of angle with respect to the trench. The lines are fits to a cosine squared function. The degree of polarization is estimated by $(I_{max} - I_{min})/(I_{max} + I_{min})$, where $I_{ax}$ ($I_{min}$) is the maximum (minimum) PL emission intensity. The excitation energy is adjusted to $E_{22}$ of 1.653 eV with a power of 5 µW. **b** PL decay curves taken from the CNT/WSe$_2$ heterostructures for $E_{11}$ (blue dots) and $IX_2$ (red dots). Shortpass (1.033 eV) and longpass filters (0.919 eV) are used for the measurements of $E_{11}$ and $IX_2$, respectively. The pulsed laser is used here with the power adjusted to 2 nW, and the excitation energy is 1.653 eV. The gray curves are the instrument response function (IRF), and the black curves are the exponential fitting curves convoluted with the IRF. **c** PL spectra at powers of 0.04, 0.4, and 4 µW from bottom to top. The excitation energy is 1.680 eV. **d** Integrated PL intensity as a function of the laser power for $E_{11}$ (blue), $IX_1$ (green), and $IX_2$ (red), respectively.

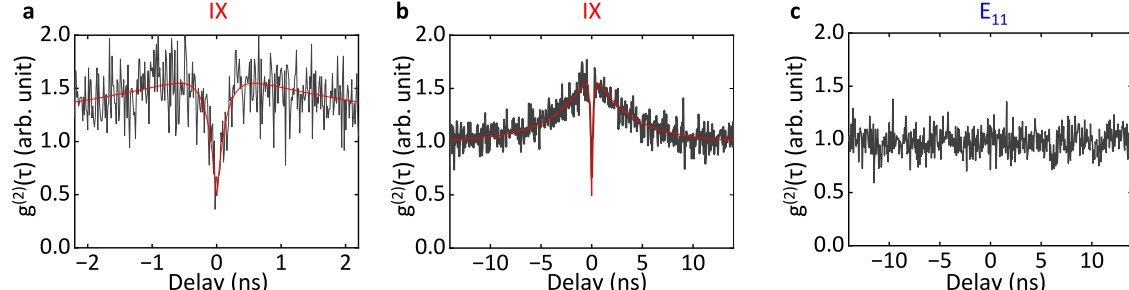

**Fig. 4 | Room-temperature quantum emission from low-energy interface excitons. a–c** Second-order correlation statistics of the IX peak during **a** short and **b** long time scales, and of **c** the $E_{11}$ peak for the (9,4) CNT/2L WSe$_2$ sample. The excitation energy is adjusted to $E_{22}$ of 1.653 eV with a power of 2 µW. Shortpass (1.033 eV) and longpass filters (0.886 eV) are used for the measurements of $E_{11}$ and IX, respectively. The gray lines are experimental results, and the red lines are the fittings. See Supplementary Note 14 for the background correction procedure used in **a**, **b**. For the second-correlation statistics data in **b**, **c**, four data points are binned together to reduce the noise. See Supplementary Note 15 and 16 for additional data.

the interface excitons. The localization is also supported by the observed blinking noise from the unstable IXs that host only "on" and "off" states (see Supplementary Figs. 3 and 13). The pure two-level noise is exclusively observed in quasi-0D systems, such as quantum dots and single molecules[31,32], suggesting that the interface exciton state could function as a single-photon emitter.

**Room-temperature quantum emission from interface excitons**
To elucidate the quantum nature of interface excitons, we conduct photon correlation measurements on a (9,4) CNT/2L WSe$_2$ sample under continuous-wave laser excitation. Two primary peaks are observed in this sample, originating from the stable low-energy IX and $E_{11}$ (see Supplementary Figs. 3 and 14). The second-order correlation $g^{(2)}(\tau)$ from the IX is shown in Fig. 4a, after background correction. A distinct antibunching dip is observed at zero delay with $g^{(2)}(0) = 0.467$.

Over a longer timescale, we also observe a bunching peak (Fig. 4b). The bunching behavior is often observed in single-photon sources and is associated with the dynamics between excited states and other dark, charged, or meta-stable states[33,34]. We employ the equation $g^{(2)}(\tau) = [1 - \alpha \exp(-|\tau|/\tau_A)] * [1 + \beta \exp(-|\tau|/\tau_B)]$ to fit the observed $g^{(2)}(\tau)$ statistics, where factors $\alpha$ and $\beta$ quantify the degree of antibunching and bunching with values of 0.720 and 0.666, respectively[35]. $\tau_A$ and $\tau_B$ indicate the timescales of the antibunching dip and the bunching peak, with values of 0.148 ns and 3.752 ns from the fitting,

respectively. The value of $1 - \alpha$ is 0.280, revealing high single-photon purity by considering the effect of bunching behavior. It is noteworthy that most of the low-energy interface excitons display similar peak linewidths and power saturation behavior (see Supplementary Fig. 12), implying that each of them acts as a quantum emitter. This is further supported by the high reproducibility of the antibunching behavior in other samples (see Supplementary Figs. 15 and 16). It is noted that an electron in CNT would sit at the Γ point in the CNT Brillouin zone, and a hole will be at the K point and the Γ point in monolayer and 2-4L WSe$_2$, respectively[15]. The corresponding interface excitons therefore could be momentum indirect. However, the strong localization observed would relax the momentum selection rule, potentially explaining the bright PL emission from the interface excitons.

For comparison, $g^{(2)}(\tau)$ from $E_{11}$ excitons does not exhibit any antibunching or bunching behavior, as shown in Fig. 4c. Under pulsed laser excitation, the $E_{11}$ excitons are known to go through an efficient exciton-exciton annihilation (EEA) process that could result in SPE[36]. The exciton density is generally lower with continuous-wave excitation, hindering the SPE through EEA. The confinement effect is crucial for SPE, and the absence of any antibunching behavior therefore indicates the 1D free feature of $E_{11}$ excitons.

We now discuss the possible origins of the localized interface exciton states, which would require a potential depth exceeding a few multiples of the thermal energy. The confinement can be provided by

defect states, for example in materials such as diamond[37], silicon carbide[38], hexagonal boron nitride[34], and CNTs[39]. It is unlikely that defects in CNTs play a role, since we use pristine suspended CNTs containing negligible exciton quenching sites[11,36]. We note that they have been characterized under low-power conditions within dry nitrogen gas environment, precluding the formation of defects in the CNTs[11]. In comparison, WSe$_2$ flakes inherently encompass a range of defect states, spanning from single vacancies to complex vacancy clusters[40,41]. Among them, it is known that single tungsten vacancies induce defects states close to the valance band[41], which could be responsible for the confinement. The defect-bound interface excitons may be formed, similar to the case of interlayer excitons in 2D-2D heterostructures[42]. The variability of the emission energy is consistent with the picture in which defects play a role, as the location and the geometry of the defect with respect to the CNT can influence the excitonic states.

As another possible explanation, inhomogeneous strain could also contribute to the localization of interface excitons as in the case of monolayer WSe$_2$[43]. The morphology of the heterostructure is examined with an atomic force microscope (Supplementary Fig. 17), and a clean CNT/WSe$_2$ interface is confirmed which would facilitate the formation of interface excitons. In one heterostructure, we observe a shallow local dip with a depth of ~3 nm and a width of ~350 nm, which may confine interface excitons nearby. Such nanoscale strain might also impact the sample through various mechanisms from van der Waals gap fluctuation[44] to lattice reconstruction[45], in a manner similar to 2D-2D heterostructures. While the exact impact of strain remains unclear, the spatially indirect nature renders interface excitons more susceptible to the aforementioned effects than intralayer excitons.

In conclusion, we observe numerous IX peaks at the CNT/WSe$_2$ interface below the CNT E$_{11}$ energy, spanning the telecommunication wavelengths. By systematically varying the chirality of CNTs and the layer number of WSe$_2$, we are able to assign the peaks to interface excitons as they only appear for type-II band alignment. The low saturation power and the long lifetime indicate that low-energy interface excitons are strongly confined, and photon antibunching is confirmed. The observation of interface excitons as room-temperature quantum emitters at telecommunication bands opens up new opportunities for applications in quantum technologies and optoelectronics, underscoring the emerging potential of mixed-dimensional heterostructures.

## Methods

### Air-suspended carbon nanotubes
We prepare air-suspended CNTs using trenched SiO$_2$/Si substrates[11]. First, we pattern alignment markers and trenches with lengths of 900 μm and widths ranging from 0.5 to 3.0 μm onto the Si substrates using electron-beam lithography, followed by dry etching. We then thermally oxidize the substrate to form a SiO$_2$ film, with a thickness ranging from 60 to 70 nm. Another electron-beam lithography process is used to define catalyst regions along the edges of the trenches. A 1.5 Å thick iron (Fe) film is deposited as a catalyst for CNT growth using an electron beam evaporator. CNTs are synthesized by alcohol chemical vapor deposition at 800 °C for 1 min. The Fe film thickness is optimized to control the yield for preparing isolated CNTs. The PL images and PL polarization measurements are performed to exclude the existence of any quenching sites in the CNTs, and only the tubes showing a smooth profile along the length and a high polarization degree > 90% are selected and used for the preparation of the heterostructures[11,12,36]. We select isolated, fully suspended chirality-identified CNTs with lengths ranging from 0.5 to 2.0 μm to form the heterostructures with WSe$_2$.

### Anthracene crystal growth
For transferring WSe$_2$ flakes onto CNTs, we grow anthracene crystals through an in-air sublimation process[17,18]. Anthracene powder is heated to 80 °C on a glass slide, while another glass slide is placed 1 mm above the anthracene source. Thin and large-area single crystals are then grown on the glass surface. To promote the growth of large-area single crystals, we pattern the glass slides using ink from commercial markers. The typical growth time for anthracene crystals is 10 h.

### Transfer of WSe$_2$ by anthracene crystals
First, WSe$_2$ (HQ graphene) flakes are prepared on 90-nm-thick SiO$_2$/Si substrates using mechanical exfoliation, and the layer number is determined by optical contrast. An anthracene single crystal is picked up with a glass-supported PDMS sheet to form an anthracene/PDMS stamp. Next, the WSe$_2$ flakes are picked up by pressing the anthracene/PDMS stamp against a substrate with the target WSe$_2$ flakes. The stamp is quickly separated ( > 10 mm/s) to ensure that the anthracene crystal remains attached to the PDMS sheet. The stamp is then applied to the receiving substrate with the desired chirality-identified CNT, whose position is determined by a prior measurement. Precise position alignment is accomplished with the aid of markers prepared on the substrate. By slowly peeling off the PDMS ( < 0.2 μm/s), the anthracene crystal with the WSe$_2$ flake is released on the receiving substrate. Sublimation of anthracene in air at 110 °C for 10 min removes the anthracene crystal, leaving behind a clean suspended CNT/WSe$_2$ heterostructure. This all-dry process eliminates contamination from solvents, and the solid single-crystal anthracene protects the 2D flakes and the CNT during the transfer, ensuring that the CNT/WSe$_2$ heterostructure experiences minimal strain[17,18].

### PL measurements
A homebuilt confocal microscopy system is employed to perform PL measurements for interface excitons and E$_{11}$ excitons at room temperature in dry nitrogen gas[11,14]. We utilize a wavelength-tunable continuous-wave Ti:sapphire laser for excitation, with its power controlled by neutral density filters. The excitation polarization angle is adjusted to be parallel to the CNT axis and the emission polarization angle dependence is measured using a half-wave plate followed by a polarizer placed in front of a spectrometer. The laser beam is focused on the samples with an objective lens that has a numerical aperture of 0.65 and a working distance of 4.5 mm. The $1/e^2$ spot diameter and the collection spot size defined by a confocal pinhole are approximately 1.2 and 5.4 μm, respectively. PL is collected through the same objective lens and detected using a liquid-nitrogen-cooled 1024-pixel InGaAs diode array connected to the spectrometer. A 150-lines/mm grating is used to obtain a dispersion of 0.52 nm/pixel at a wavelength of 1340 nm. For photoluminescence measurements of WSe$_2$ A excitons, a 532-nm laser and a charge-coupled device camera are employed.

### Time-resolved and photon correlation measurements
Approximately 100 femtosecond pulses at a repetition rate of 76 MHz from a Ti:sapphire laser is utilized for time-resolved measurements. The excitation laser beam is focused onto the sample using an objective lens with a numerical aperture of 0.85 and a working distance of 1.48 mm. The PL from the center of the nanotube within the heterostructure is coupled to a superconducting single-photon detector with an optical fiber, and a time-correlated single-photon counting module is used to collect the data. IRFs dependent on the detection wavelength are acquired by dispersing supercontinuum white light pulses with a spectrometer. Photon correlation measurements are carried out using a Hanbury-Brown-Twiss setup with a 50:50 fiber coupler under excitation with a continuous-wave laser. The experiments are conducted at room temperature.

## Data availability
All the data generated in this study have been deposited into the R2DMS-GakuNinRDM database, and are accessible at https://dmsgrdm.riken.jp/42tsf/.

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

## Acknowledgements

Parts of this study are supported by JSPS (KAKENHI JP22K14624 to D.Y., JP22K14625 to S.F., JP21K14484 to M.M., JP22K14623 to C.F.F., JP22H01893 to D.K., JP21H05233 to S.O., JP23H00262, JP22F22350, JP20H02558 to Y.K.K.) and MEXT (ARIM JPMXP1222UT1135). Y.R.C. is supported by JSPS (International Research Fellow). N.F. and C.F.F. are supported by RIKEN Special Postdoctoral Researcher Program. We thank the Advanced Manufacturing Support Team at RIKEN for technical assistance.

## Author contributions

N.F. carried out sample preparation and performed measurements on the samples. Y.R.C. performed atomic force microscope measurements and assisted in sample preparation. Y.R.C., C.F.F., and K.N. assisted in optical measurements. D.Y., S.F., and D.K. contributed to the time-resolved PL and photon correlation measurements. M. M., Y.G., and S.O. performed density functional theory calculations. K.O. aided in the development of the anthracene-assisted dry transfer method. Y.K.K.

supervised the project. N.F. and Y.K.K. co-wrote the manuscript, with all authors providing input and comments on the manuscript.

## Competing interests

The authors declare no competing interests.
