## [Peer Review File · Nature Communications]

Room-temperature quantum emission from interface excitons
in mixed-dimensional heterostructuresREVIEWER COMMENTS

Reviewer #1 (Remarks to the Author):

Fang et al. studied emission properties of heterostructures consisting of 2D WSe₂ and 1D carbon nanotubes. By stacking the two types of materials on top of each other over trenches, low energy emission peaks were observed, which they assigned to interlayer excitons. These peaks exhibited a certain localization nature at room temperature, manifested as pump-dependent emission saturation and long lifetimes.

The heterostructures studied here can potentially reveal interesting new physics due to the different dimensionalities and band structures of the two types of materials. The manuscript is well written, and the data are clearly presented. That being said, several of the major conclusions lack clear experimental support, or are in contradiction with their data. Due to these reasons, I would not recommend the publication of this manuscript unless the authors can address these serious issues.

1. The very claim of interlayer excitons. The authors assigned the low energy peaks from the heterostructures to interlayer excitons. When stacking a 1D and a 2D semiconductor together, if a type-II band offset is formed, it should result in a mostly 1D like interlayer exciton feature (electrons in nanotubes, and holes in WSe₂). Instead, distinct peaks associated with localized excitons were observed. These observations are suggestive of defect related localized states, which are quite common in TMDs. The authors should perform systematic control experiments to elucidate the contributions of TMD defects. These include, e.g. recording spectra of suspected TMDs in the low energy ranges, before and after being stacked with nanotubes.

2. The authors should provide statistics of IX peak positions, in particular for the 1L WSe₂ samples. Are the peak positions of the IXs reproducible from tube to tube? This information could help reveal the origins of these low-energy peaks.

3. The authors use the disappearance of the low energy peaks in small bandgap nanotube/WSe₂ heterostructures as supporting evidence for their assignment of the interlayer excitons. However, this argument is difficult to justify given that the spectral range where the interlayer excitons are expected to emerge are barely covered in the detection window (e.g. from (12,4) to (10,8)). Also, as the authors displayed in fig. 3c, the intensity ratio between the nanotube emission and the low energy peak depends a lot on the excitation power. Experimental data that considers both detection range and excitation power should be included to strengthen the argument.

4. For ease of comparison, the authors should label the bandgap of WSe₂ 1L/.3L in Fig. 2b.

5. Morphology of the heterostructures. The way that the samples were prepared can easily cause the WSe₂/nanotube heterostructures to bend towards and even touch the trench bottom. The resultant strains can easily lead to localized defect emission from the WSe₂. The authors should perform careful

structural characterization to understand the morphologies of the heterostructures and the influence of defects.

6. Another important aspect is the interface between the WSe₂ and the nanotubes. To form interlayer excitons, charge carriers need to tunnel from one material to the other. This requires the interface between the 1D and 2D materials to be free of debris and ultraclean. The weak emission from the low energy peaks, if assigned to interlayer excitons, suggests inefficient charge tunneling.

7. Polarization data. Interlayer excitons have their charge carriers separated in the top WSe₂ and bottom nanotube. As such, its polarization should reflect a vertical dipole, rather than an in-plane one that follows the orientation of the nanotube. The authors should explain this contradictory phenomenon.

8. Second-order correlation data. The smallest g_2 value that the authors have presented is 0.467, which is close to the 0.5 value typically used for justifying the observation of single photon emission. More examples with $g_2 \ll 0.5$ should be provided in order for the authors to claim single photon emission. Otherwise, I would recommend dropping such claims.

Reviewer #2 (Remarks to the Author):

Dear Editor,

The manuscript reports on the observation of indirect excitons (IXs) at the 2D-WSe₂/CNT (2D/1D) interface at room temperature. They incorporated steady-state PL, polarization-dependent PL, lifetime PL, and photon-correlation measurements.

Steady-state PL measurements demonstrated the formation of two new emission peaks at the 2D/1D interface (Fig. 1). The authors assigned these new peaks IXs. To prove this assignment, the authors conducted control experiments, which consisted of controlling the "IX" emission by running the band-alignment of the interface through the CNT chirality (Fig. 2). This is an excellent way for confirming the hypothesis that observed new emission peaks at the heterostructure originate from IXs. Furthermore, the authors conducted photon correlation measurements (Fig. 4) to check the single photon emission. Although the observation of IX formation at heterostructures involving 2D TMDs is not new, the manuscript brings in my opinion to main new elements about IXs in heterostructures involving 2D-TMDs: 1) the extension of exciton lifetime (Fig. 3b), making excitons that live longer is one of the main goals of making these heterostructures, and 2) proving the single photon emission, which may be useful in incorporating these hybrids in quantum computing and information technologies.

The manuscript is well written, and the results are well presented and interpreted. In my opinion, the manuscript can be published in Nature Comm. as is.

Reviewer #3 (Remarks to the Author):

Review of Fang et al.,

In this work, the authors, by creating suspended heterostructures of WSe₂-SWCNTs, observe single photon emission lines below the E₁₁ transition of SWCNTs. These SPEs are attributed to the interlayer(interface) exciton between WSe₂ and CNTs, with the hole residing in the WSe₂ and the electron in the CNT. However, in my opinion, the experimental evidence is not sufficient to prove these SPEs are interlayer excitons and to rule out other possibilities, such as just single-photon defects in the CNT itself.

Moreover, interlayer defect-bound exciton SPEs have been previously observed in WSe₂-MoS₂ heterostructures[1], so it is plausible that similar exciton complexes could be observed in WSe₂-CNTs heterostructures. Therefore, for this paper to distinguish itself and be novel enough to merit publication in Nature Communications, I would have expected a lot more analysis and insights into the nature of the SPEs and why, unlike other interlayer excitons in the 2D heterostructures, they can operate at room temperature. Also, assessing the utility of these SPEs, previously telecom band single photon emitters with room temperature operation have been identified in defects in CNTs and also separately in TMD (MoTe₂-albeit at cryogenic temperature). Given that the $g_2(0)$ of these emitters hardly reaches 0.5 and their cryogenic linewidth and brightness are not assessed, the utility of these SPEs is also questionable. Given these, I cannot recommend this work for publication in its current form; a lot more experiments and analysis are needed for this work to be suitable for nature communication publication. I'll discuss my main concerns below:

1- The IX₁ and IX₂ are very similar to previously observed sp-3 defect doublets (E₁₁^{*}, E₁₁^{*-}) in CNTs. They both tend to appear as doublets, are in the same spectral range, and have similar lifetimes. I understand that authors screen the CNT before transfer to make sure there are no defects in the CNT; however, it is plausible that the transfer process itself could either cause damage to the CNT or trap some adsorbates at the interface that may lead to defect creation. Even in Fig.1d (PL before the transfer), I noticed a small signal close to 1.03 eV, which, given the background noise level, could have been resolved better with higher integration time and averaging. Furthermore, could the WSe₂ could activate dim sp-3 defects in the CNT by providing a larger absorption cross-section and exciton funneling. Neither of these two possibilities was discussed at length in the text, and providing some controlled experiments that could rule out such scenarios (specially the first one) is critical to the paper's claims.

2- In the explanation of Fig.1i, the authors state, "E₁₁ shows a strong response to excitation energy corresponding to the CNT E₂₂ transition. A similar response as E₁₁ is observed for IX₁ and IX₂ peaks, implying that the carriers forming the IXs are supplied from the CNT. In comparison, we do not observe a clear signature of the WSe₂ A exciton peak in the PLE map." This statement needs to be clarified. The PLE peak at (x=1.1eV, y~1.675) aligns perfectly with the A exciton of WSe₂, but the authors attribute it to the E₂₂ excitation of the CNT. In fact, in your recent publication [2] in Fig.1d, you show a PLE map of a CNT-WSe₂ heterostructure, and you assign the peak at the exact location (1.673 eV) to the WSe₂ exciton transfer process. In literature, the E₂₂ transition for (9,4) CNTs is reported ~1.72 eV [3-4]. So, I cannot follow how you justify that carriers are supplied from CNT. More PLE maps from different sizes CNT/1L

WSe2 are needed. Also, it would be helpful to perform the PLE down to E11 excitation energies.

3- In Fig.2b, as E11 is red-shifted, the measurement background noise is also increasing. The counts of the Sp-3 defects in CNT also tend to decrease as they get red-shifted [5] due to decreased confinement energy. Because the SPEs are becoming dimmer and the PL background noise is increasing, it is likely that the SPEs exist and cannot be resolved within this limit. A great way to increase the confidence in the data is to do cryogenic PL, where the emitters should become much brighter and discernable. Then, a similar data set from Fig.2e can be interpreted more confidently.

4- The band alignments in Fig.2a and 2c need to be quantified, preferably using rigorous DFT simulations. What is the energy difference between the WSe2 valance/conduction band and the CNT valance/conduction bands? How does CNT/WSe2 band alignment evolve as a function of different CNT chiralities? For instance, as E11 is increased for smaller diameter CNTs, how much is the contribution of the conduction band moving up in energy, and how much is the valance band moving down? Does the spectral shift of the SPE quantitatively follow the shift in the valence band energy of WSe2 as Wse2 layers are increased? Using ab initio methods or modeling, you could predict your expected energy of the interlayer exciton for each configuration and see how well the spectral position of the SPEs lines up with the predicted interlayer band alignment. This could give you an estimate for the energy position of any defects that could have localized this exciton similarly to [1].

5- From a bandstructure perspective, an electron in CNT would sit at gamma point, and a hole in monolayer WSe2 would be at K point. Authors discussed the spatially indirect dipole would dim the transition, but this transition is also momentum indirect. I was expecting a discussion on how this would still lead to observable emissions. Also, for bilayer WSe2, the valence band at the gamma point moves up to almost equal energy as the K-point, and then for trilayer and more, the gamma point should sit higher than the K-point in WSe2. If the hole is residing in the WSe2 valance band, then you should be able to see some drastic effects on your PL emission of the interlayer excitons, given the change in the momentum of the holes as you go from 1L to 3L. Do you observe any signature of this transition?

6- Ideally, a CNT/Wse2 should lead only to a 1d confinement. To further confine these excitons into 0D, authors mention defects or strain could play a role. However, the authors should discuss the possible origins of these emission lines more in-depth and more analysis to back up their claims. For instance, selenium and Sulphur defects in WSe2 and MoS2 are mostly electron trapping, and in the case of previously observed defect-bound interlayer excitons in WSe2-MoS2 heterostructures, the defects in MoS2 are responsible for trapping the electron while the hole resides in the band edge. In the case of CNT/WSe2 heterostructure, the proposed band alignment shows that electrons are in the CNT, so I don't think it would be plausible for single or double selenium defects in WSe2 to play a role, if it's a defect in WSe2, then it should be a hole trapping defect. A discussion along these lines and whether a potential suitable defect in WSe2 exist that could have the right energy for band-alignment is needed.

Also, again given that the emission lines appear as doublets, similar to sp-3 defect lines in CNT, it is highly plausible that either the electron's or the hole's wavefunction(or both) is bound to a defect on the CNT, which can have two distinct binding configurations and give rise to a doublet structure. If no defect in CNT itself is responsible for the emission and the electron resides in the CNT band-edge, then authors

should discuss why these emission lines even appear as doublets and not just a single emission line, as previously observed in WSe₂-MoS₂ heterostructures.

7- angled SEM images of the final device are needed to get an idea of the strain fields and also possible damages of the transfer process to the CNT. Is the WSe₂ draping over both the CNT and the trench?

[1]- arXiv:2205.02472

[2]- arXiv:2307.07124

[3]- Wei, Xiaojun, et al. "Experimental determination of excitonic band structures of single-walled carbon nanotubes using circular dichroism spectra." *Nature communications* 7.1 (2016): 12899.

[4]- Podlesny, Blazej, et al. "En route to single-step, two-phase purification of carbon nanotubes facilitated by high-throughput spectroscopy." *Scientific Reports* 11.1 (2021): 10618.

[5]- He, Xiaowei, et al. "Tunable room-temperature single-photon emission at telecom wavelengths from sp³ defects in carbon nanotubes." *Nature Photonics* 11.9 (2017): 577-582.

Response to reviewer report for manuscript No. NCOMMS-23-41893-T by Fang *et al.*

We thank the reviewers for taking their time to review the manuscript and providing helpful comments. Our point-by-point response and revisions made are provided below.

Response to Reviewer #1:

Fang et al. studied emission properties of heterostructures consisting of 2D WSe₂ and 1D carbon nanotubes. By stacking the two types of materials on top of each other over trenches, low energy emission peaks were observed, which they assigned to interlayer excitons. These peaks exhibited a certain localization nature at room temperature, manifested as pump-dependent emission saturation and long lifetimes.

The heterostructures studied here can potentially reveal interesting new physics due to the different dimensionalities and band structures of the two types of materials. The manuscript is well written, and the data are clearly presented. That being said, several of the major conclusions lack clear experimental support, or are in contradiction with their data. Due to these reasons, I would not recommend the publication of this manuscript unless the authors can address these serious issues.

We appreciate the positive comments, in particular, stating that our study “can potentially reveal interesting new physics” and noting that “The manuscript is well written, and the data are clearly presented”. We have made a major revision of the manuscript to address the reviewer’s concerns. Below we provide detailed point-by-point response to the reviewer’s comments.

1. The very claim of interlayer excitons. The authors assigned the low energy peaks from the heterostructures to interlayer excitons. When stacking a 1D and a 2D semiconductor together, if a type-II band offset is formed, it should result in a mostly 1D like interlayer exciton feature (electrons in nanotubes, and holes in WSe₂). Instead, distinct peaks associated with localized excitons were observed. These observations are suggestive of defect related localized states, which are quite common in TMDs. The authors should perform systematic control experiments to elucidate the contributions of TMD defects. These include, e.g. recording spectra of suspected TMDs in the low energy ranges, before and after being stacked with nanotubes.

We completely agree that 1D-2D heterostructures should result in 1D-like interface excitons, and we expected to see delocalized 1D excitonic states before performing these experiments. Instead, we observed localization and single-photon emission at room temperature, which is quite surprising and interesting. Although the energy range of 0.79-1.07 eV for the IX peaks is considerably lower than the energy range of 1.50-1.70 eV for known defect-bound exciton emission from WSe₂ (A. Ripin *et al.*, Nat. Nanotechnol. 18, 1020, 2023; K. Parto *et al.*, Nat. Commun. 12, 3585, 2021), we also agree with the reviewer’s point that emission from defects in the WSe₂ flake should be ruled out. In fact, in the last paragraph on Page 5, we have provided spatial and spectral overlap between E₁₁ and the IX peak as experimental evidence that IX peaks originate from the interface.

Nevertheless, we provide additional data which correspond to the control experiment suggested by the reviewer. In our 1D-2D heterostructures, the WSe₂ flakes are typically larger than 10 μm and therefore a region spatially displaced from the CNT would serve as a control. In Fig. R1, the results of hyperspectral PL imaging of the (9,4) CNT/1L WSe₂ heterostructure in Fig. 1 are summarized. The six panels on the left show PL spectra at different locations indicated in the top PL image on the right. P1 is the location of the (9,4) CNT, exhibiting three IX peaks labeled as IX_{R1}, IX_{R2}, and IX_{R3}. P5 and P6 correspond to spectra from suspended WSe₂, where no IX peaks are observed. If the IX peaks originate from defects in WSe₂, we expect similar emission peaks uniformly distributed over the flake. PL images at the energies of the IX peaks are shown on the right, indicating no other emitters within the image. We have also performed similar measurements on 3 samples, and in all cases the IX peaks overlap spatially with the CNT emission.

Fig. R1 (left) PL spectra taken at six positions, indicated by blue circles as P1, P2, P3, P4, P5, and P6. The sharp peaks at 0.93 eV in spectra from P3 and P4 are from another (9,7) CNT. (right) The PL intensity maps of E₁₁, IX_{R1}, IX_{R2}, and IX_{R3} for the (9,4) CNT/1L WSe₂ heterostructure as shown in Fig. 1 in the main text. The scale bars are 1 μm . The apparent sizes of the emission spots differ because of the strong saturation behavior of the interface excitons combined with the Airy pattern. The excitation energy is 1.653 eV and the excitation power is 10 μW .

To clarify this point, we have added Fig.R1 as Supplementary Fig. 4 and have added a sentence to the last paragraph of Page 5 (Lines 106-108):

“Such spatial overlap with the CNT emission indicates that IX peaks cannot be explained by emission from randomly distributed defect states within WSe₂.”

2. The authors should provide statistics of IX peak positions, in particular for the 1L WSe₂ samples. Are the peak positions of the IXs reproducible from tube to tube? This information could help reveal the origins of these low-energy peaks.

We appreciate the reviewer’s comment regarding the statistics. Since there are only four 1L samples out of which only one showing the IX peaks, we present a histogram in Fig. R2 that describes the statistical information of IX peak energies for all samples. We observed a broad range of IX peaks energies from 0.79 to 1.07 eV, randomly distributed over our detection window. The energy dispersion cannot be attributed to the varying chiralities of the CNTs, because peak positions and number of the peaks are different even for heterostructures with identical chirality and layer number as shown in Fig. R3 in response to Question 3 below.

Fig. R2 Histogram of IX peak energies from all the samples.

To provide detailed information, we have added Fig. R3 as Supplementary Fig. 7 and also added Supplementary Table 1 to list all the prepared samples.

Regarding the statistics for 1L heterostructures, we agree that it may simplify the analysis. However, preparing a single heterostructure is laborious, since identification of an appropriate pristine CNT is required and a WSe₂ flake needs to be transferred onto a trench precisely at the selected CNT location. To collect data from the 35 samples in this manuscript, we spent 13 months. Monolayer flakes are especially fragile, resulting in a low yield of less than 20%. Given the difficulty of the experiment, we wish to publish with the current data set to initiate discussion within the community.

3. The authors use the disappearance of the low energy peaks in small bandgap nanotube/WSe₂ heterostructures as supporting evidence for their assignment of the interlayer excitons. However, this argument is difficult to justify given that the spectral range where the interlayer excitons are expected to emerge are barely covered in the detection window (e.g. from (12,4) to (10,8)). Also, as the authors displayed in fig. 3c, the intensity ratio between the nanotube emission and the low energy peak depends a lot on the excitation power. Experimental data that considers both detection range and excitation power should be included to strengthen the argument.

We thank the reviewer for commenting on an important point. First, to consider the detection range, we have plotted $\Delta E = E_{11} - E_{IX}$ versus E_{11} for the IX peaks as shown in Fig. R3, where the black line shows the detector limit. For CNTs exhibiting IX peaks, the highest-energy IX peaks observed are close to the E_{11} energies with $\Delta E < 0.09$ eV, and most of them are located around $\Delta E \approx 0.05$ eV. Although the reviewer is correct in pointing out that the identification of the IX peaks is limited by the detection range for E_{11} energies below 0.80 eV, the high-energy IX peaks for CNTs with E_{11} energies of 0.85–0.95 eV would be within the detection window. The absence of such IX peaks for CNTs with $E_{11} < 0.94$ eV therefore supports our interpretation by band alignment transition from type-II to type-I.

Fig. R3 IX peak energies. ΔE plotted as a function of E_{11} energy from 35 samples. The black line indicates the detection limit of the detector. Inverted triangles, upright triangles, circles, and rectangles indicate 1L, 2L, 3L, and 4L WSe₂ samples, respectively. For the samples absent of IX peaks, we plot the symbols at $\Delta E = 0$ eV.

To include this discussion, we have added Fig. R3 in Supplementary Note 7, and we refer to this note in the main text on Page 8, Line 143.

Next we consider the power dependence. As the reviewer commented, the ratio between E_{11} and IX emissions is sensitive to excitation power, making it difficult to identify the IX peaks for higher powers when PL spectra are normalized. This is caused by the saturation of the IX peaks, but the absolute PL intensity of these peaks monotonically increases with increasing power, as depicted in Fig. 3c,d and Supplementary Fig. 10. Hence the spectra at higher powers are advantageous for identification of the IX peaks. We note that most spectra in Fig. 2b were taken at $10 \mu\text{W}$ where IX peak height should be saturated. To identify the existence of the IX peaks, we have replotted the spectra from several samples in Fig. 2b with a zoomed-in vertical axis as Fig. R4. For (14,0) CNT/4L WSe₂, (8,7) CNT/2L WSe₂, (12,1) CNT/2L WSe₂, and (9,4) CNT/2L WSe₂ heterostructures, the IX peaks can be identified with peak counts larger than 20 counts/sec. For (15,1) CNT/3L WSe₂, (9,8) CNT/1L WSe₂, (12,4) CNT/3L WSe₂, and (13,2) CNT/3L WSe₂ heterostructures, we should be able to detect any peaks higher than 10 counts/sec considering the noise floor. We do not observe any additional peaks, which again supports our interpretation with band alignment transition.

We therefore have added Fig. R4 as Supplementary Fig. 6 and this figure has been referred to in the main text on Page 6, Line 134.

Fig. R4 Zoomed-in PL spectra from different heterostructures shown in Fig. 2b. Excitation energy is adjusted to E_{22} for each heterostructure. Excitation power values are 4, 6, and $5 \mu\text{W}$ for (9,4)/2L, (12,1)/2L, (8,7)/2L heterostructures, respectively, and $10 \mu\text{W}$ for other samples. Weak peaks at $\sim 0.14 \text{ eV}$ below the E_{11} states are the K-momentum states E_K , whose assignment is well established in the literature (Please also see our response to Question 1, Reviewer #3).

4. For ease of comparison, the authors should label the bandgap of WSe₂ 1L/..3L in Fig. 2b.

We tried to label the bandgap of WSe₂, but the values are outside the energy range of the spectra plotted in Fig. 2b. The exciton energies of 1L to 4L WSe₂ ranges from 1.66 eV to 1.44 eV as shown in Supplementary Fig. 2, whereas the spectra in Fig. 2b shows emission in the range from 0.77 eV to 1.15 eV.

To clarify the bandgap of the WSe₂ flakes, we refer to Supplementary Fig. 2 on Page 6, Line 118.

5. Morphology of the heterostructures. The way that the samples were prepared can easily cause the WSe₂/nanotube heterostructures to bend towards and even touch the trench bottom. The resultant strains can easily lead to localized defect emission from the WSe₂. The authors should perform careful structural characterization to understand the morphologies of the heterostructures and the influence of defects.

We sincerely thank the reviewer for an important comment. As requested, we have conducted AFM measurements to characterize the morphology of our CNT/WSe₂ heterostructures, and the AFM images are illustrated in Fig. R5. Two CNTs can be seen over the trench, as marked by arrows. The WSe₂ flake is draped over both the CNT and the trench. As the reviewer commented, the suspended part falls into the trench by approximately 6 nm. We note that the trench depth is over 2.2 μm , and therefore the suspended heterostructure would not touch the bottom of the trench.

Fig. R5 (left, top) An optical microscope image of a CNT/2L WSe₂ heterostructure. 2D (left, bottom) and 3D (right) AFM images for the sample. The inset in (left, bottom) is a line profile indicated by the green broken line. The scale bars are 5 and 0.4 μm for the left, top and left, bottom images, respectively.

Regarding the influence of defects, localized defect emission from the WSe₂ should be in the energy range of 1.50-1.70 eV. Since the IX peaks have much lower energies of 0.79-1.07 eV, it is unlikely that they originate from defects in WSe₂. In addition, the observed band alignment transition cannot be explained by localized defect emission from WSe₂.

We therefore have added Fig. R5 as Supplementary Fig. 14 which has been referred to in the third paragraph on Page 12.

6. Another important aspect is the interface between the WSe₂ and the nanotubes. To form interlayer excitons, charge carriers need to tunnel from one material to the other. This requires the interface between the 1D and 2D materials to be free of debris and ultraclean. The weak emission from the low energy peaks, if assigned to interlayer excitons, suggests inefficient charge tunneling.

We agree with the reviewer that the interface is important. As the reviewer commented, the formation of interface (interlayer) excitons requires the interface between 1D and 2D materials to be free of debris and ultraclean. To obtain such a clean surface, we used the special anthracene-assisted dry transfer technique (K. Otsuka, *et al.* Nat. Commun. 12, 3138, 2021). Since anthracene can be sublimed at a low temperature of 110°C, the surface can be kept clean of residues. In our response to the previous question, AFM image is shown as Fig. R5. The suspended part of the WSe₂ flake exhibits a uniformity characterized by a reduced surface roughness, highlighting the cleanliness of our transfer process.

We also agree with the reviewer that the formation of interface excitons requires charge carriers to tunnel from one material to the other. We recently reported exciton transfer in CNT/WSe₂ heterostructures but with type-I band alignment (arXiv:2307.07124), providing experimental evidence for charge carrier tunneling. It is therefore reasonable that charge transfer also occurs in CNT/WSe₂ heterostructures with type-II band alignment.

Regarding the intensity of the IX peaks, high excitation powers were used for Fig. 2b, resulting in saturation of IX peaks and low relative intensity with respect to the E₁₁ emission. The intrinsic intensities should be compared at low excitation powers where emission scales linearly. As shown in Fig. 3c, the IX peaks can be brighter than E₁₁ peak at low excitation powers, suggesting sufficient charge tunneling to form interface excitons with efficiencies comparable to E₁₁ excitons.

To clarify the cleanness of the interface and the formation process of interface excitons, we have revised the main text as follows:

On Page 5, the first paragraph, we have added a sentence: “The formation of the indicated indirect excitons generally requires charge transfer, which is plausible as exciton transfer has been observed in similar heterostructures with type-I band alignment [15].”

On page 12, the third paragraph, we have added a sentence: “The morphology of the

heterostructure is examined with an atomic force microscope (Supplementary Fig. 14), and a clean CNT/WSe₂ interface is confirmed which would facilitate the formation of interface excitons.”

7. Polarization data. Interlayer excitons have their charge carriers separated in the top WSe₂ and bottom nanotube. As such, its polarization should reflect a vertical dipole, rather than an in-plane one that follows the orientation of the nanotube. The authors should explain this contradictory phenomenon.

We thank the reviewer for the insightful comment. We also expected interface excitons to have a vertical dipole, and we were surprised to see linear polarization dependence. However, this does not preclude the presence of an out-of-plane dipole component as well. Due to the use of a conventional normal-incidence photoluminescence setup which predominantly detects in-plane dipoles, the vertical component remains unresolved in our experiments.

We speculate that the linear polarization observed in interface excitons arise from the one-dimensional nature of electrons within CNTs. The band alignment studies in the third and fourth paragraphs on Page 6 suggest that electrons are predominantly within CNTs while holes are located in WSe₂. Given the inherent 1D nature of electrons in CNTs, the movement of electrons is constrained along the 1D axis. In such mixed-dimensional systems, a complex scenario involving both vertical and in-plane dipoles may be able to explain the experimental observations.

There are, however, further complications due to the nontrivial behavior of the polarization dependence. A noticeable deviation in emission polarization angle from E₁₁ is observed as demonstrated in Fig. 3a and Supplementary Fig. 9. The emission angle deviation shows variations, ranging from 7° to 21° across three samples, suggesting that the polarization is not solely determined by electron motion within the CNT. The angular disparity has been reported in other localized systems (Y. Bai, *et al.*, Nat. Mater., 19, 1068, 2020; N. R. Jungwirth and G. D. Fuchs, Phys. Rev. Lett., 119, 057401, 2017), possibly stemming from many-body interactions or external factors like local strain. More research for a definitive explanation is needed which is beyond the scope of our current manuscript, and we refrain from speculating further at present.

To include the above discussion, we have added a paragraph in Supplementary Note 9.

To raise the issue of the out-of-plane dipole, we have added a sentence in the third paragraph on Page 8 as: “Owing to the use of a conventional normal-incidence photoluminescence setup which predominantly detects in-plane dipoles, the vertical component remains unresolved in our experiments.”

8. Second-order correlation data. The smallest g₂ value that the authors have presented is 0.467, which is close to the 0.5 value typically used for justifying the observation of single photon emission. More examples with g₂ <<0.5 should be provided in order for the authors to claim single photon emission. Otherwise, I would recommend dropping such claims.

We agree that demonstrating reproducibility is important. We would like to bring your attention the data shown in Supplementary Note 13, which is shown below as Fig. R6. The lowest g⁽²⁾(0)

that we achieved from an IX peak in a (12,0) CNT/2L WSe₂ heterostructure reaches 0.33, sufficiently lower than 0.5. We note that the values of $1 - \alpha$, which reflect the single-photon purity considering the effect of bunching, are 0.280 and 0.110 for Fig. 4b and Fig. R6 (right), respectively.

Fig. R6 (left) A PL spectrum from the (12,0) CNT/2L WSe₂ heterostructure. Peaks at 0.811 and 1.026 eV correspond to IX₇₋₁ and E₁₁, respectively. (right) Second-order correlation statistics of IX₇₋₁. The excitation energy is adjusted to 1.531 eV of E₂₂ with a continuous-wave laser power of 0.5 μW. A longpass filter (0.826 eV) is used to collect PL emission from IX₇₋₁. The red line is the fitting.

Reviewer #2 (Remarks to the Author):

Dear Editor,

The manuscript reports on the observation of indirect excitons (IXs) at the 2D-WSe₂/CNT (2D/1D) interface at room temperature. They incorporated steady-state PL, polarization-dependent PL, lifetime PL, and photon-correlation measurements.

Steady-state PL measurements demonstrated the formation of two new emission peaks at the 2D/1D interface (Fig. 1). The authors assigned these new peaks IXs. To prove this assignment, the authors conducted control experiments, which consisted of controlling the “IX” emission by running the band-alignment of the interface through the CNT chirality (Fig. 2). This is an excellent way for confirming the hypothesis that observed new emission peaks at the heterostructure originate from IXs. Furthermore, the authors conducted photon correlation measurements (Fig. 4) to check the single photon emission. Although the observation of IX formation at heterostructures involving 2D TMDs is not new, the manuscript brings in my opinion to main new elements about IXs in heterostructures involving 2D-TMDs: 1) the extension of

exciton lifetime (Fig. 3b), making excitons that live longer is one of the main goals of making these heterostructures, and 2) proving the single photon emission, which may be useful in incorporating these hybrids in quantum computing and information technologies. The manuscript is well written, and the results are well presented and interpreted. In my opinion, the manuscript can be published in Nature Comm. as is.

We sincerely thank the reviewer for concisely summarizing and describing the novelty of our results. In particular, we are encouraged by the reviewer's assessment as "This is an excellent way for confirming the hypothesis" and the comment on the novelty of this work as "main new elements about IXs in heterostructures", and we are happy to see that the reviewer suggests publication in Nature Communications as is.

Reviewer #3 (Remarks to the Author):

Review of Fang et al.,

In this work, the authors, by creating suspended heterostructures of WSe₂-SWCNTs, observe single photon emission lines below the E₁₁ transition of SWCNTs. These SPEs are attributed to the interlayer(interface) exciton between WSe₂ and CNTs, with the hole residing in the WSe₂ and the electron in the CNT.

However, in my opinion, the experimental evidence is not sufficient to prove these SPEs are interlayer excitons and to rule out other possibilities, such as just single-photon defects in the CNT itself. Moreover, interlayer defect-bound exciton SPEs have been previously observed in WSe₂-MoS₂ heterostructures[1], so it is plausible that similar exciton complexes could be observed in WSe₂-CNTs heterostructures. Therefore, for this paper to distinguish itself and be novel enough to merit publication in Nature Communications, I would have expected a lot more analysis and insights into the nature of the SPEs and why, unlike other interlayer excitons in the 2D heterostructures, they can operate at room temperature. Also, assessing the utility of these SPEs, previously telecom band single photon emitters with room temperature operation have been identified in defects in CNTs and also separately in TMD (MoTe₂-albeit at cryogenic temperature). Given that the g₂(0) of these emitters hardly reaches 0.5 and their cryogenic linewidth and brightness are not assessed, the utility of these SPEs is also questionable. Given these, I cannot recommend this work for publication in its current form; a lot more experiments and analysis are needed for this work to be suitable for nature communication publication. I'll discuss my main concerns below:

We are grateful for the reviewer's thorough effort for preparing this report regarding our experimental observation of localized interface excitons in WSe₂-SWCNTs heterostructures and encouraging us toward publication. We have made substantial changes by providing many more experimental results and analysis to address the reviewer's concerns. The point-by-point response to the comments is shown below.

1- The IX1 and IX2 are very similar to previously observed sp³ defect doublets (E11,E11*-) in CNTs. They both tend to appear as doublets, are in the same spectral range, and have similar lifetimes. I understand that authors screen the CNT before transfer to make sure there are no defects in the CNT; however, it is plausible that the transfer process itself could either cause damage to the CNT or trap some adsorbates at the interface that may lead to defect creation. Even in Fig.1d (PL before the transfer), I noticed a small signal close to 1.03 eV, which, given the background noise level, could have been resolved better with higher integration time and averaging. Furthermore, could the WSe₂ could activate dim sp³ defects in the CNT by providing a larger absorption cross-section and exciton funneling. Neither of these two possibilities was discussed at length in the text, and providing some controlled experiments that could rule out such scenarios (specially the first one) is critical to the paper's claims.*

We thank the reviewer for bringing up an important point. We agree that some aspects of the IX peaks are similar to sp³ defects in CNTs, but there are distinct differences in the experimental observations. Below we describe the differences and consider the scenario suggested by the reviewer.

First, we generally observed many IX peaks simultaneously, which is different from the doublets observed for sp³ defects. As illustrated in Supplementary Fig. 3a,b, eight IX peaks are identified from one (9,4) CNT/2L WSe₂ heterostructure sample. Second, these peaks display abrupt blinking where many of them are uncorrelated, diverging from the behavior of sp³ defects which exhibit much more stable emission (X. He *et al.* Nat. Photonics, 11, 577, 2017; Y. Piao *et al.* Nat. Chem., 5, 840, 2013). Third, we observe an abrupt transition for the appearance of IX peaks with CNT E₁₁ energy, indicating the importance of band alignment as discussed in the response for Reviewer #1, Question 3. For sp³ defects, there exists no such transition and any chirality CNT can form sp³ defects (D. Kozawa *et al.* Nat. Commun., 13, 2814, 2022). From these experimentally observed differences, we feel that it is difficult to assign these peaks to sp³ defects.

To include the above discussion, we have revised the text as shown below:

On Page 5, Line 92 in the second paragraph, we have added the following sentence: “Such PL evolution is not observed in sp³ defects, which exhibit more stable emission once formed [22,23].”

On Page 6, Line 129 in the third paragraph, we have added the following sentence: “It is noted that sp³ defects in CNTs generally introduce doublet peaks [23], different from the numerous IX peaks observed here (see Supplementary Fig. 3).”

On Page 6, Line 136 in the third paragraph, we have also added the following sentence: “We note that for sp³ defects, there exists no such transition and any chirality CNT can form sp³ defects [25].”

Nevertheless, we consider the two possibilities that the reviewer suggested. We start with the possibility of dim sp³ defects existing prior to heterostructure formation and being activated by

WSe₂.

Of the heterostructures exhibiting IX peaks in Fig. 2b, PL spectra of pristine CNTs before heterostructure formation are shown in Fig. R7. All these tubes show a main peak from E₁₁ states with high PL intensities. Weak peaks at ~0.14 eV below the E₁₁ states denoted as E_K are the K-momentum states, whose assignment is well established in the literature (O. Torrens *et al.*, Phys. Rev. Lett. 101, 157401, 2008; P. M. Vora *et al.*, Phys. Rev. B, 81, 155123, 2010; J. L. Blackburn *et al.*, Nano Lett., 1398, 12, 2012). The “small signal close to 1.03 eV” that reviewer noticed is also the E_K peak. We have excluded the E_K peaks for the analysis of the IX peaks, as noted in the caption of Fig. 2.

To clarify the absence of sp³ defects prior to heterostructure formation, we have added Fig. R7 as Supplementary Fig. 1.

On Page 3, Line 71, we have revised a sentence as: “High-quality CNTs are initially grown over trenches (see Methods and Supplementary Fig. 1)”.

A description of K-momentum peaks in the caption of Figure 1 has been added: “The peaks at 1.011 and 0.962 eV in (d) and (e) are K-momentum exciton peaks.”

Fig. R7 PL spectra of pristine (14,0), (10,5), (12,1), and (9,4) CNTs, which are used for the formation of heterostructures in Fig. 2.

Besides these E₁₁ and E_K peaks, no other peaks related with defects could be resolved. This is reasonable since we also measured the PL images and polarization for the pristine tubes, and only the tubes with a smooth profile along the length and a high polarization degree > 90 % are selected to exclude the existence of any quenching sites on the CNTs. Similar selection process has been used in our previous work, where we show that the CNTs are clean and free of defects (A. Ishii *et al.*, Phys. Rev. B, 91, 125427, 2015; A. Ishii *et al.*, Phys. Rev. Applied 8, 054039, 2017; A. Ishii *et al.*, Phys. Rev. X, 9, 041048, 2019).

We have added the above description in the paragraph for air-suspended carbon nanotubes in the Methods section as: “The PL images and PL polarization measurements are performed to exclude the existence of any quenching sites in the CNTs, and only the tubes showing a smooth profile

along the length and a high polarization degree > 90 % are selected used for the preparation of the heterostructures [11,12,36].”

Regarding sp^3 defect activation by exciton funneling from WSe_2 , we expect to observe more IX peaks when exciton transfer is efficient in such a case. We have reported that exciton transfer is only observed for type-I band alignment (arXiv:2307.07124), whereas IX peaks mainly appear in CNTs with type-II band alignment (please also see our response to Reviewer #1, Question 3 and Fig. R11). Since the observation of IX peaks is anticorrelated with the exciton transfer as also noted in the response to the next question, we feel that defect activation by exciton funneling is unlikely.

To include this point, we have modified a sentence in the first paragraph of Page 8: “Exciton transfer observed in similar heterostructures show anticorrelation with the appearance of the IX peaks, consistent with the band alignment transition (Supplementary Fig. 8) [15].”

Next, we consider the possibility of transfer induced defects. We have previously performed characterization of CNT/hBN heterostructures (N. Fang *et al.* ACS Photonics 7, 1773, 2020), which serves as a control experiment for assessing the effect of the transfer process. Figure R8 shows PL spectra from a pristine (10,5) CNT, and a CNT/hBN heterostructure fabricated from the same tube. For the heterostructure, both E_{11} and E_k peaks are redshifted due to dielectric screening by hBN. No new peaks could be resolved, indicating that the transfer process does not introduce defect states in CNTs.

We have also performed AFM measurements to characterize the morphology of the heterostructure in response to Question 5 by Reviewer #1. In Fig. R5, the suspended part of the WSe_2 flake exhibits a uniformity characterized by a reduced surface roughness, highlighting the cleanliness of our transfer process.

Based on the above discussion, we have added a description of the hBN/CNT heterostructures in Line 88, Page 5: “The transfer process is not expected to introduce defects, since CNT/hexagonal boron nitride heterostructures prepared in a similar manner does not exhibit such low-energy peaks [14].”

Fig. R8 PL spectra for a (10,5) CNT (left) before and (right) after the transfer of a hBN flake to form the heterostructure.

2- In the explanation of Fig.1i, the authors state, "E11 shows a strong response to excitation energy corresponding to the CNT E22 transition. A similar response as E11 is observed for IX1 and IX2 peaks, implying that the carriers forming the IXs are supplied from the CNT. In comparison, we do not observe a clear signature of the WSe2 A exciton peak in the PLE map.". This statement needs to be clarified. The PLE peak at $(x=1.1\text{eV}, y\sim 1.675)$ aligns perfectly with the A exciton of WSe2, but the authors attribute it to the E22 excitation of the CNT. In fact, in your recent publication [2] in Fig.1d, you show a PLE map of a CNT-WSe2 heterostructure, and you assign the peak at the exact location (1.673 eV) to the WSe2 exciton transfer process. In literature, the E22 transition for (9,4) CNTs is reported $\sim 1.72\text{ eV}$ [3-4]. So, I cannot follow how you justify that carriers are supplied from CNT. More PLE maps from different sizes CNT/1L WSe2 are needed. Also, it would be helpful to perform the PLE down to E11 excitation energies.

We thank the reviewer for the important comment. The reviewer is correct that the PLE peak in Fig. 1i is very close in energy to the A exciton of WSe₂. We can, however, distinguish the two by comparing the line shape and the excitation polarization dependence. In Fig. R9, we compare PLE spectra of E₁₁ emission for the (9,4) CNT/1L WSe₂ sample (Fig. 1i in this manuscript) and the (10,5) CNT/1L WSe₂ sample which is known to show exciton transfer (Fig. 3a in arXiv:2307.07124). Although the peaks are energetically close, the A exciton absorption peak (E_A) in the (10,5) CNT/1L WSe₂ sample is considerably broader.

Fig. R9 (left) Normalized PLE spectra of the E₁₁ emission obtained by integrating over a 30-meV-wide spectral window centered at the E₁₁ energy from (red) the (10,5) CNT/1L WSe₂ and (blue) the (9,4) CNT/1L WSe₂ samples.

In addition, excitation polarization measurements would result in strong linear polarization for the CNT E₂₂ peak, whereas WSe₂ A exciton absorption would be isotropic. The peak in Fig. 1i is linearly polarized, and we optimize the excitation polarization to obtain maximum emission intensity for the PL measurements. We have measured six heterostructures fabricated from (9,4) CNT, and such polarization optimization has been performed on all of them. Unfortunately, we do not have the full polarization dependent PL intensity data for the sample in Fig. 1i, but we present data from two (9,4) CNT/2L WSe₂ samples in Fig. R10. These two samples show polarization degree of 0.84 and 0.61.

Fig. R10 (left) Normalized PLE spectra of the E_{11} emission obtained by integrating over a 30-meV-wide spectral window centered at the E_{11} energy from the (12,4) CNT/2L WSe₂ (red) and the (9,4) CNT/2L WSe₂ samples (blue for #1 and purple for #2). (right) Excitation polarization dependence of E_{11} emission for the (12,4) CNT/2L WSe₂ sample (excitation energy: 1.65 eV), the #1 (9,4) CNT/2L WSe₂ sample (excitation energy: 1.69 eV), and the #2 (9,4) CNT/2L WSe₂ sample (excitation energy: 1.70 eV). The lines are fits to a cosine squared function.

For comparison with another sample exhibiting exciton transfer, we also show the PLE spectrum and excitation polarization dependence for the (12,4) CNT/2L WSe₂ sample in Fig. R10. The PLE spectrum shows the broad A exciton peak and the polarization is isotropic. Based on such spectral lineshape and polarization dependence observed in multiple samples, we are confident that the peak in Fig. 1i arises from the E_{22} resonance of the (9,4) CNT.

To clarify that the excitation peak in Fig. 1i originates from the E_{22} transition, we have added Fig. R10 as Supplementary Fig. 5. We have also revised a sentence in Page 5, Line 108 as: “In the PLE map (Fig. 1i), E_{11} shows a strong response to excitation energy, which is identified as CNT E_{22} transition (Supplementary Fig. 5).”

Regarding the energy of the (9,4) CNT E_{22} peak in Fig. 1i, the reviewer is correct in pointing out the difference from the literature. Our data shows peak energy of 1.675 eV, which is slightly different from the value of 1.72 eV reported in the references listed by the reviewer. These values, however, are not directly comparable as the CNTs measured in the references are surfactant encapsulated solutions. Our heterostructure samples show varying amounts of redshifts compared to air-suspended tubes due to dielectric screening as mentioned on Line 84, and this particular sample shows a large shift. We are also certain that this nanotube is (9,4) since we perform PLE measurements before transfer to determine the chirality for all samples.

To clarify this point, we have added a sentence on Page 3 Line 73: “We perform PLE measurements before transfer to determine the chirality for all samples.”

Regarding more PLE maps, as requested, we present additional PLE maps from different samples as shown in Fig. R11. (12,1) and (14,0) CNT based heterostructures do not show the E_A peak, while the new low-energy emission peaks are observed. In comparison, the other CNTs with smaller bandgap exhibit exciton transfer.

Fig. R11 PLE maps of different heterostructures. The excitation power is $5 \mu\text{W}$ for (12,1) CNT/2L WSe₂ and (14,0) CNT/4L WSe₂ samples and $10 \mu\text{W}$ for others. The excitation polarization is aligned to CNT axis.

We have added Fig. R11 as Supplementary Fig. 8 to include more PLE maps.

Regarding PLE measurement down to E_{11} excitation energies, we agree it may provide additional information about the relaxation process from the E_{11} excitons to interface excitons, but we believe that we have provided sufficient data to distinguish the CNT E_{22} transition from the WSe₂ A exciton transition. We also note that it is experimentally difficult as the energies are beyond the limit of our Ti:sapphire laser.

3- In Fig.2b, as E_{11} is red-shifted, the measurement background noise is also increasing. The counts of the Sp^3 defects in CNT also tend to decrease as they get red-shifted [5] due to decreased confinement energy. Because the SPEs are becoming dimmer and the PL background noise is increasing, it is likely that the SPEs exist and cannot be resolved within this limit. A great way to increase the confidence in the data is to do cryogenic PL, where the emitters should become much brighter and discernable. Then, a similar data set from Fig.2e can be interpreted more confidently.

We thank the reviewer for carefully checking our data. As we explained in our response to Question 1, there are distinct differences in the experimental observations between sp^3 defects and IX peaks. Nevertheless, we consider the possibility of SPEs not being resolved in these data.

We have replotted the PL spectra with a y-axis representing counts/sec as Fig. R4 in response to

Question 3 from Reviewer #1, where the background noise can be clearly seen. The noise level variation in different samples is due to different integration time during the PL measurements. The maximum standard deviation observed for background noise is 4.8 counts/sec in the (15,1) CNT/3L WSe₂ sample, and most samples have lower noise levels below 3.0 counts/sec. In comparison, the least intense IX peak still present PL intensity over 20 counts/sec, confirming that these spectra have sufficient signal to noise ratio. If these peaks are gradually becoming dimmer, we should observe lower intensity peaks. Instead, we observe a sharp transition in the disappearance of IX peaks as discussed in our response to Question 3 from Reviewer #1.

We would also like to note that the PL background noise is not necessarily increasing with decreasing E_{11} . For example, in Fig. 2b, the noise levels in (12,4) CNT/3L WSe₂ and (9,8) CNT/1L WSe₂ samples lower, while becoming high in (9,7) CNT/3L sample.

To show the noise level in the PL spectra, we have added Fig. R4 as Supplementary Fig. 6.

Regarding cryogenic PL measurements, we agree that the interface excitons may become brighter, but we already have sufficient signal to detect the IX peaks as discussed above. It is certainly a good suggestion to perform cryogenic measurements as they will provide additional important information on the interface excitons. Our current manuscript, however, focuses on reporting the observation of interface excitons at room temperature, and we feel that such measurements are beyond the scope of this paper.

4- The band alignments in Fig.2a and 2c need to be quantified, preferably using rigorous DFT simulations. What is the energy difference between the WSe₂ valance/conduction band and the CNT valance/conduction bands? How does CNT/WSe₂ band alignment evolve as a function of different CNT chiralities? For instance, as E_{11} is increased for smaller diameter CNTs, how much is the contribution of the conduction band moving up in energy, and how much is the valance band moving down? Does the spectral shift of the SPE quantitatively follow the shift in the valence band energy of WSe₂ as Wse₂ layers are increased? Using ab initio methods or modeling, you could predict your expected energy of the interlayer exciton for each configuration and see how well the spectral position of the SPEs lines up with the predicted interlayer band alignment. This could give you an estimate for the energy position of any defects that could have localized this exciton similarly to [1].

We appreciate the reviewer's suggestion. As requested, we have used DFT simulations to examine the band alignment in this system as shown in Fig. R12. The simulated band structures involve various zigzag CNTs and 1-4L WSe₂, qualitatively consistent with the experimental results as discussed below. These DFT simulation results are included as Supplementary Figure 6 in our recent work (arXiv:2307.07124).

Fig. R12 Band alignment of CNT/WSe₂ heterostructures from DFT calculations. The calculated electronic band structures of (13,0), (10,0) and (8,0) CNT as well as 1L, 2L, 3L, and 4L WSe₂. The energies are measured from the vacuum level.

For a CNT with a small bandgap, such as the (13,0) CNT shown here, a type-I band alignment is formed in the heterostructure. With increasing the CNT bandgap (or increasing E_{11} energy for smaller diameter CNTs) through chirality alteration from (13,0) to (8,0), valance band moves down and the transition of band alignment occurs. Qualitatively, the transition from type-I to type-II with increasing CNT bandgap is consistent with Fig. 2b. Regarding the layer number dependence of WSe₂, 1L WSe₂ forms type-I band alignment with (8,0) CNT, while 4L WSe₂ forms type-II, again qualitatively explaining the layer number dependence shown in Fig. 2d.

Using the simulation results to answer the reviewer's questions, the energy differences between the valance bands and the conduction bands are 0.57 eV and 0.65 eV, respectively, for (13,0) CNT/1L WSe₂ heterostructure. As the conduction band minimum of CNT stays within the bandgap of WSe₂ regardless of the diameter, the band alignment is determined by the behavior of valance band only. The valance band of WSe₂ increases by 0.13 eV from bilayer to quadlayer, while the IX peaks are broadly distributed over 0.3 eV (Fig. R3) and quantitative comparison is difficult.

Although one may argue that a more accurate picture will be provided by DFT simulations with the exact heterostructures perfectly reproducing the experiments, this is computationally not trivial. Many CNT chiralities involve much more than 100 atoms in a unit cell. For example, the (9,4) and (12,1) CNTs, which exhibit many IX peaks, involve 532 and 628 C atoms in a unit cell, respectively. Furthermore, the lattice constant for WSe₂ is considerably different from CNT. Because integer multiples of a WSe₂ unit cell will not equal the size of a CNT unit cell, impractically large unit cell size will be required. The computational resources needed to run such simulations will be unrealistically enormous. Alternatively, strain can be imposed in one of the materials to accommodate the different sizes of the unit cells, but then the simulation will no longer be reproducing the experiments.

Accordingly, we have revised one sentence in the third paragraph of Page 6 to refer to the data as follows:

“The presence of the IX peaks is determined by chirality, consistent with the transition in band alignment from type-II to type-I that is observed in a density functional theory simulation [15].”

5- From a band structure perspective, an electron in CNT would sit at gamma point, and a hole in monolayer WSe₂ would be at K point. Authors discussed the spatially indirect dipole would dim the transition, but this transition is also momentum indirect. I was expecting a discussion on how this would still lead to observable emissions. Also, for bilayer WSe₂, the valence band at the gamma point moves up to almost equal energy as the K-point, and then for trilayer and more, the gamma point should sit higher than the K-point in WSe₂. If the hole is residing in the WSe₂ valence band, then you should be able to see some drastic effects on your PL emission of the interlayer excitons, given the change in the momentum of the holes as you go from 1L to 3L. Do you observe any signature of this transition?

We are grateful for the insightful comments. Regarding the transition over the layer numbers, we do not observe drastic effects on PL emission of the IXs for 1L compared to 2L. The PL spectra from the (9,4) CNT-based samples with WSe₂ layer number from 1L to 3L are shown in Fig. R13. Both 2L samples show IX peaks with comparable intensity with the 1L sample. For the 3L sample, the IX peaks show somewhat weaker emission, but not as dramatic as that expected from the above scenario. In addition, Fig. 2d show PL spectra from the (10,5) CNT-based samples with WSe₂ layer number from 1L to 4L. Although the 1L sample does not show any IX peaks, the changes with the layer number are not drastic but rather gradual.

To include the above discussion, we have added a sentence on Line 125, Page 6 as: “It is noted that the valence band maximum of WSe₂ changes from the K point to the Γ point with increasing the layer number, but correlation with the behavior of IX peaks cannot be observed.”

Fig. R13 PL spectra for the (9,4) CNT-based heterostructures with different WSe₂ layer number. The excitation power is 4 μ W and the excitation polarization is aligned to CNT axis for all the samples. Two different samples are prepared for the 2L WSe₂, which are labelled as #1 and #2 in Question 2. The peak located at 0.91 eV for the (9,4) CNT/3L WSe₂ sample comes from E₁₁ emission from another tube.

Regarding the comment on momentum conservation, we agree with the reviewer that an electron in CNT would sit at the Γ point in the CNT Brillouin zone, and a hole will be at the K point and

the Γ point in monolayer and 2-4L WSe₂, respectively. In consideration of the momentum selection rules, the 1D CNT Brillouin zone needs to be mapped onto the 2D Brillouin zone. As shown in the Fig. 14 (left), the Γ point of the lowest conduction band in the 1D CNT Brillouin zone is located close to the K point for the 2D graphene Brillouin zone. We therefore need to consider the overlap of graphene K point with WSe₂ K and Γ points. We plot the extended Brillouin zone of graphene and WSe₂ in Fig. R14 (right). The lattice constants for graphene and WSe₂ are 0.246 and 0.328 nm, respectively, resulting in no apparent periodicity for a general stacking angle. As a result, the graphene K point samples the WSe₂ Brillouin zone in a quasi-uniform manner. Somewhere in the extended Brillouin zone, the graphene K point can overlap with the WSe₂ K and Γ points where the momentum selection rule can be satisfied. In addition, the observed interface excitons are highly localized, implying that the momentum selection rules would be relaxed. In either case, the transition becomes allowed, explaining the observation of emission from interface excitons.

Fig. R14 (left) Brillouin zone of a CNT mapped onto the graphene Brillouin zone. Cutting lines for CNT are shown. (right) Extended Brillouin zone for graphene and WSe₂.

Based on the above discussion, on the first paragraph, Page 11, we have added the following sentences: “It is noted that an electron in CNT would sit at the Γ point in the CNT Brillouin zone, and a hole will be at the K point and the Γ point in monolayer and 2-4L WSe₂, respectively [15]. The corresponding interface excitons therefore should be momentum indirect. However, the strong localization observed would relax the momentum selection rule, potentially explaining the bright PL emission from the interface excitons.”

6- Ideally, a CNT/WSe₂ should lead only to a 1d confinement. To further confine these excitons into 0D, authors mention defects or strain could play a role. However, the authors should discuss the possible origins of these emission lines more in-depth and more analysis to back up their claims. For instance, selenium and Sulphur defects in WSe₂ and MoS₂ are mostly electron trapping, and in the case of previously observed defect-bound interlayer excitons in WSe₂-MoS₂ heterostructures, the defects in MoS₂ are responsible for trapping the electron while the hole resides in the band edge. In the case of CNT/WSe₂ heterostructure, the proposed band alignment shows that electrons are in the CNT, so I don't think it would be plausible for single or double selenium defects in WSe₂ to play a role, if it's a defect in WSe₂, then it should be a hole trapping defect. A discussion along these lines and whether a potential suitable defect in WSe₂ exist that could have the right energy for band-alignment is needed.

Also, again given that the emission lines appear as doublets, similar to sp³ defect lines in CNT, it is highly plausible that either the electron's or the hole's wavefunction(or both) is bound to a defect on the CNT, which can have two distinct binding configurations and give rise to a doublet structure. If no defect in CNT itself is responsible for the emission and the electron resides in the CNT band-edge, then authors should discuss why these emission lines even appear as doublets and not just a single emission line, as previously observed in WSe₂-MoS₂ heterostructures.

We thank the reviewer for the comments discussing the origin of the IX peaks. As commented by the reviewer, we agree that selenium defects cannot be responsible for the formation of localized interface excitons, since they are electron trapping defects. As a potential candidate, single tungsten vacancies have been reported to introduce defect states close to valance band (S. Zhang, *et al.*, Phys. Rev. Lett. 119, 046101, 2017). The defect-bound interface excitons therefore are possible, similar to reference [1] as the reviewer commented.

Regarding why the emission does not appear as a single emission line, we indeed observe many IX peaks simultaneously, which is different from the doublets observed for sp³ defects as addressed in response to Question 1. The variability of the emission energy is consistent with the picture in which defects play a role, as the location and the geometry of the defect with respect to the CNT can influence the excitonic states.

To clarify the exact defect type in WSe₂, we have revised the manuscript in the second paragraph of Page 12 and referred to reference [1] as follows:

“WSe₂ flakes inherently encompass a range of defect states, spanning from single vacancies to complex vacancy clusters [40, 41]. Among them, it is known that single tungsten vacancies induce defects states close to the valance band [41], which could be responsible for the confinement. The defect-bound interface excitons may be formed, similar to the case of interlayer excitons in 2D-2D heterostructures [42]. The variability of the emission energy is consistent with the picture in which defects play a role, as the location and the geometry of the defect with respect to the CNT can influence the excitonic states.”

In the same paragraph, we have deleted the following sentence: “Although most defects in WSe₂ yield shallow energy traps and exhibit SPE only under cryogenic conditions, these defects may be responsible for the trapping potential for the interface excitons.”

7- angled SEM images of the final device are needed to get an idea of the strain fields and also possible damages of the transfer process to the CNT. Is the WSe₂ draping over both the CNT and the trench?

We appreciate the reviewer's suggestion. We have conducted SEM measurements on the CNT/WSe₂ heterostructures. In the SEM images, a single suspended CNT is generally difficult to be identified and bundles are more readily resolved. As shown in Fig. R15, a Y-junction CNT bundle was observed in a pristine sample. For the heterostructure sample, the CNTs beneath the WSe₂ flake was not observed although we carefully scanned the sample under SEM.

We therefore present AFM images instead of the angled SEM to characterize the morphology (please see Fig. R5 in response to Question 5 for Reviewer #1). Two CNTs are discernible across the trench, as marked by arrows. The WSe₂ flake is draped over both the CNT and the trench. The tubes do not exhibit apparent damages from the transfer process. Instead, we observed a local strain in the upper tube, as indicated by a green arrow in Fig. R5 (right), which may confine interface excitons. We therefore have added Fig. R5 as Supplementary Fig. 14 and have revised the third paragraph in Page 12 as follows:

“In one heterostructure, we observe a shallow local dip with a depth of ~3 nm and a width of ~350 nm, which may confine interface excitons nearby. Such nanoscale strain might also impact the sample through various mechanisms from van der Waals gap fluctuation [44] to lattice reconstruction [45], in a manner similar to 2D-2D heterostructures.”

Fig. R15 SEM images of a suspended CNTs (left) and a CNT/3L WSe₂ heterostructure (middle, right). The white circle in the middle image indicates the measured area for the right image. The scale bars are 2, 2, 1.5 μm , respectively, from left to right.

[1]- [arXiv:2205.02472](https://arxiv.org/abs/2205.02472)

[2]- [arXiv:2307.07124](https://arxiv.org/abs/2307.07124)

[3]- Wei, Xiaojun, et al. "Experimental determination of excitonic band structures of single-walled carbon nanotubes using circular dichroism spectra." *Nature communications* 7.1 (2016): 12899.

[4]- Podlesny, Blazej, et al. "En route to single-step, two-phase purification of carbon nanotubes facilitated by high-throughput spectroscopy." *Scientific Reports* 11.1 (2021): 10618.

[5]- He, Xiaowei, et al. "Tunable room-temperature single-photon emission at telecom wavelengths from sp³ defects in carbon nanotubes." *Nature Photonics* 11.9 (2017): 577-582.

We thank the reviewer for providing relevant references. We cite the above references [1,2,5] as Refs. [42,15,23] in the manuscript, respectively, where we have updated the citations to arXiv papers as they are both published.

We hope that the reviewer is satisfied with the above response and the changes we made. Although some aspects of these emitters are not completely understood, they are novel emitter states found in the mixed-dimensional heterostructures. Compared to 2D-2D interlayer excitons, not only are the dimensionality different but they operate at room temperature. We feel that publication of our unexpected experimental results will stimulate discussion and give inspiration for theories addressing the remaining details.

REVIEWER COMMENTS

Reviewer #1 (Remarks to the Author):

I appreciate the authors' efforts in addressing some of my concerns. However, I retain my recommendation that its publication in Nature Communications in its current form is not appropriate based on two main reasons: (1) Their claims still lack strong and convincing experimental data support, and (2) the lack of in-depth discussions about the very nature of the emission peaks. Please note that I consider the major novelty of the work to be from point (2), namely in-depth discussions about the origin of the so-claimed interlayer emission, which could only be derived from and reversely consolidate the experimental conclusions in point (1). In their manuscript, I feel the claim of RT interlayer single photon emission still lacks convincing experimental data support, and there is barely any discussion about the nature of the associated electronic states. I therefore could not recommend its publication in Nature Communications.

- The claim of lack of IX in small bandgap tubes is not convincing because the detection window does not cover the relevant energy range (Fig S8). Unless the authors can provide convincing evidence that the binding energies of these new excitonic states are so small that they are fully covered by the detection window, the authors should consider rephrasing or removing these data points, and they should not be used as evidence for supporting their IX claims.

- I'm not convinced by the authors' explanation about the linear polarization observed for the low energy peaks. As mentioned earlier, if it is indeed from localized IX, an out of plane dipole rather than an in-plane one should be expected. The dimensionality of the carrier hosts should not affect this much. The observation of the predominant in plane linear polarization reflects that most of the photons in the emission spectra using the normal incident geometry are from in-plane recombination. This is in contradiction with the very claim that these peaks originate from IXs.

- Based on the authors' response to my comment 2 in the previous revision, I'm a bit concerned about the repeatability of their observations. It is mentioned that the red shifted peaks are observed in only one 1L sample (I'm singling out the 1L sample because I think it's the most straightforward system to focus on, as I detail in the next point). In this case, it would be the most valuable if details about this sample, including morphology (AFM) of the final device and control measurements before and after deposition, are included in the manuscript. Most importantly, repeatability of their observation should be addressed and, in the ideal case, confirmed.

- The authors treated the 1L-4L samples on equal footing, but as reviewer 3 pointed out, band structures and carrier dynamics in 1L WSe₂ are quite different from those in thicker films. I strongly feel that the authors should consider performing more careful theoretical calculations to understand the electronic structures of these heterostructures. This could help prevent any misassignment and improve the depth of the manuscript.

- With majority of the raw data showing a $g_2 \geq 0.5$, I find the claim of antibunching and single photon emission lack convincing data support. I would recommend the authors providing g_2 data with more apparent single photon emission evidence, or removing this conclusion.

- There is a lack of discussion about the origin of these possible IX species at the fundamental level. As mentioned earlier, the potential novelty of this work lies in the origin of these interlayer states. The authors should consider combining their experimental data with detailed theoretical calculations to provide more physical insights, including band structure and electronic structure of the localized IX states.

- It seems that the current work is a follow up of their previous publication based on the same heterostructure (arXiv:2307.07124). The authors should state clearly how this work distinguishes from their previous publication.

Reviewer #3 (Remarks to the Author):

Fang et al. rereview,

The authors provided a strong rebuttal with additional data. They have addressed the significant concerns. I believe the manuscript is sufficiently revised and can be considered for publication. My final request is for the authors to provide $g_2(0)$ data on long-time scales (preferentially plot the time axis on a logarithmic scale) up to micro/mili seconds (similar to Fig.3 in ref.1). This will not only ensure the autocorrelation function is normalized to 1 on long-time delays accurately and provides a better fit for your g_2 values (for instance, plot R6.right might imply there is bunching at -4,+4 seconds and seems like an odd stopping point to normalize the g_2) but also gives very valuable information for theorists to try to model the SPE and the involved level structure.

[1] Patel, Raj N., et al. "Probing the optical dynamics of quantum emitters in hexagonal boron nitride." PRX Quantum 3.3 (2022): 030331.

Response to reviewer report for manuscript No. NCOMMS-23-41893A by Fang *et al.*

We thank the reviewers for taking their time to go through our response and providing helpful comments. Our point-by-point response and revisions made are provided below.

Response to Reviewer #1:

I appreciate the authors' efforts in addressing some of my concerns. However, I retain my recommendation that its publication in Nature Communications in its current form is not appropriate based on two main reasons: (1) Their claims still lack strong and convincing experimental data support, and (2) the lack of in-depth discussions about the very nature of the emission peaks. Please note that I consider the major novelty of the work to be from point (2), namely in-depth discussions about the origin of the so-claimed interlayer emission, which could only be derived from and reversely consolidate the experimental conclusions in point (1). In their manuscript, I feel the claim of RT interlayer single photon emission still lacks convincing experimental data support, and there is barely any discussion about the nature of the associated electronic states. I therefore could not recommend its publication in Nature Communications.

We sincerely thank the reviewer for stating his main concerns, which we have addressed to improve the manuscript. Specifically, we have performed additional experiments to strengthen our conclusions involving the preparation of new samples, measurement of their PL spectra, second-order correlations, and AFM morphology. Based on the experimental observations and DFT simulations, discussion about the electronic states and the localization of interface excitons has also been added. Below we provide detailed point-by-point response to the reviewer's comments.

- The claim of lack of IX in small bandgap tubes is not convincing because the detection window does not cover the relevant energy range (Fig S8). Unless the authors can provide convincing evidence that the binding energies of these new excitonic states are so small that they are fully covered by the detection window, the authors should consider rephrasing or removing these data points, and they should not be used as evidence for supporting their IX claims.

We appreciate the continued discussion on the detection window. We agree that the binding energies in the excitonic states would affect their emission energies and that the binding energies are large in these systems; typically 320 meV (A. Chernikov *et al.*, Phys. Rev. Lett. 113, 076802, 2014) for TMDC monolayers and 400 meV (F. Wang *et al.*, Science 308, 838, 2005) for CNTs. Even with the large binding energies, however, the interface excitons are expected to emerge close to E_{11} states in our 1D-2D heterostructures. In a simplified picture, the excitonic energy can be estimated by subtracting the binding energy of the exciton from the free carrier energy, and is represented as $E_{\text{exciton}} = E_{\text{free carrier}} - E_{\text{binding}}$. The free carrier energy can be identified in the band diagram as shown in Fig. R1. In our previous work focusing on the exciton transfer process (arXiv:2307.07124), the band alignment transition occurs through changing CNT chiralities. For a CNT with a small bandgap, a type-I band alignment is formed in the heterostructure. With increasing the CNT bandgap, valence band moves down and the transition of band alignment

occurs. Since the type-II samples investigated in this manuscript are close to the transition, the band offset in the valence band should be small. Hence $E_{\text{free carrier}}$ for interface excitons should be just slightly lower than that for CNT E_{11} excitons. Regarding the binding energies for interface excitons, we expect they are relatively large as the reviewer asserts. For suspended MoSe_2 - WSe_2 heterostructures, binding energies in interlayer excitons are around 250 meV and are comparable with intralayer excitons (S. Ovesen et al., *Commun. Phys.* 2, 23, 2019). An analogous scenario is plausible in our 1D-2D heterostructures, in consideration of comparable dielectric screening effects. The binding energy is likely comparable with CNT E_{11} excitons and this is consistent with the bright emission from interface excitons at room temperature and high PL intensity comparable with E_{11} excitons as shown in Fig. 3c. Since both free carrier energies and binding energies are similar, the total excitonic energies for interface excitons are expected to be very close to E_{11} excitons.

Fig. R1. Band diagram for type-I and type-II 1D-2D heterostructures.

In response to Comment 3 in the last round and as detailed in Supplementary Fig. 7 (Supplementary Fig. 8 in this revision), our experimental findings reveal that the highest-energy IX peaks (IX_{high}) across most samples closely approach the E_{11} energies, with a typical energy difference near 0.05 eV. To illustrate the positions of the IX_{high} peaks in each heterostructure, we plot representative PL spectra in Fig. R2. The recurrent appearance of such highest-energy IX peaks in large bandgap tubes is in line with the free carrier energies and binding energies being close in E_{11} and IXs. This characteristic allows IX_{high} peaks to act as indicators for the presence or absence of interface excitons and the “relevant energy range” can be defined as the 0.05 eV region below E_{11} . Given that our detection limit is located at 0.773 eV, it follows that IX_{high} in CNTs with E_{11} energies exceeding 0.823 eV should fall within our detection window. Furthermore, the average linewidth of IX peaks from 10 samples is approximately 0.036 eV. With this broad linewidth, the presence or absence of the IX peaks can still be determined even when the E_{11} energy is near 0.823 eV. Although we initially included a data point at 0.805 eV in Fig. 2e as the tail of the IX peak should still be detectable, we removed this data point to fulfill the rigorous criteria described above.

Fig. R2. Zoomed-in PL spectra from different heterostructures exhibiting IX_{high} peaks.

To clearly show the absence of IX_{high} peaks for CNTs with $0.823 \text{ eV} < E_{11} < 0.945 \text{ eV}$, the representative PL spectra in small bandgap tubes are shown in Fig. R3. We have also illustrated the anticipated positions of the IX_{high} peaks, should they have been present. Here, we have included PL spectra extracted from the right four panels in Supplementary Fig. 9 (Supplementary Fig. 8 in the last round) and they all cover the 0.05 eV region below E_{11} (Fig. R3, bottom four panels). The confirmed absence of IX_{high} peaks in these samples is indicative of a transition from type-II to type-I band alignment. Regarding the low-energy IX peaks, we agree with the reviewer that the current detection window may not cover all of them, as some are already close to the detection limit. The energies of these low-energy peaks are more scattered in different samples as shown in Supplementary Fig. 8, possibly coming from different trapping potential depth in each individual localized state. However, this should not affect the assessment on the absence of IXs using IX_{high} as the indicator.

Fig. R3. Zoomed-in PL spectra from different heterostructures absent of IX peaks. Red peaks and lines indicate expected IX_{high} and their center positions, respectively. The bottom four panels are extracted from the PLE maps in the right four panels in Supplementary Fig. 9 (Supplementary Fig. 8 in the last round).

To clarify the above points, we have revised two sentences and added a sentence in the third paragraph of Page 6 as shown below:

“The observed IX peaks span a broad energy range within the telecommunication wavelengths. Note that the low-energy peaks approach the edge of the spectral window, suggesting the possibility of lower-energy IX peaks existing beyond our current detection capability. Meanwhile, the highest energy peak in each chirality is located close to E_{11} , with a difference of ~ 0.05 eV.”

- I'm not convinced by the authors' explanation about the linear polarization observed for the low energy peaks. As mentioned earlier, if it is indeed from localized IX, an out of plane dipole rather than an in-plane one should be expected. The dimensionality of the carrier hosts should not affect this much. The observation of the predominant in plane linear polarization reflects that most of the photons in the emission spectra using the normal incident geometry are from in-plane recombination. This is in contradiction with the very claim that these peaks originate from IXs.

We thank the reviewer for the continued discussion on the polarization. As noted in our previous response to Comment 7 in the last round, we also expected interface excitons to have a vertical dipole. However, we would like to bring to your attention that interlayer excitons in 2D-2D heterostructures are not exclusively out-of-plane polarized and exceptions exist. For example, under one-dimensional moiré patterns, these interlayer excitons show strong in-plane linear

polarization with a high polarization degree over 90% (Y. Bai, et al., Nat. Mater. 19, 1068, 2020; S. Zhao, et al., Nat. Nanotechnol. 18, 572, 2023). It indicates that even vertically spatially-separated excitons could indeed have some characteristics of in-plane 1D excitons, depending on the dimensionality of the structures and the confinement.

A similar mechanism may be giving rise to mixed in-plane and out-of-plane polarization in our 1D-2D interface excitons, which would correspond to a tilted dipole. In such a case, observation of in-plane polarization does not rule out the existence of interface excitons. There should also be out-of-plane polarized emission, but unfortunately, as noted in the response to Comment 7 in the last round, the use of normal incidence geometry is incapable of detecting the out-of-plane component. Since the fractions of in-plane and out-of-plane components cannot be determined, we do not claim “predominant in plane linear polarization” but carefully state that “the vertical component remains unresolved”. Ideally, three-dimensional characterization of polarization would clarify the situation, but in general, this is experimentally difficult in high-resolution microscopy. In the above mentioned two papers, the out-of-plane components are also not characterized.

If the tilting of the dipole is caused by the structural anisotropy and/or strain, the direction of the tilt is not necessarily along the nanotube axis. The polarization deviation observed in Fig. 3a and Supplementary Fig. 11 is consistent with such a picture as discussed in the previous response. In addition, we have performed additional experiments and found even more drastic polarization deviation. In a recently prepared (9,4) CNT/2L WSe₂ sample, we observed an intriguing IX peak, which is labeled as IX_{R4-2} in Fig. R5. Distinct from other observed IX peaks such as IX_{R4-1} and IX_{R4-3}, IX_{R4-2} displays an almost perpendicular angle misalignment with the E₁₁ exciton. Moreover, we observed a remarkably lower in-plane linear polarization degree of 0.41 compared with other IX peaks.

To include the above discussion and Fig. R5, we have revised Supplementary Note 11.

Fig. R5. (left) A PL spectrum from the (9,4) CNT/2L WSe₂ heterostructure. (right) Emission polarization dependence of PL emission from E₁₁ (blue circles), IX_{R4-1} (orange circles), IX_{R4-2} (red circles), and IX_{R4-3} (wine circles). The lines are fits to a cosine squared function. The excitation energy is 1.703 eV and excitation power is 10 μW.

- Based on the authors' response to my comment 2 in the previous revision, I'm a bit concerned about the repeatability of their observations. It is mentioned that the red shifted peaks are observed in only one 1L sample (I'm singling out the 1L sample because I think it's the most straightforward system to focus on, as I detail in the next point). In this case, it would be the most valuable if details about this sample, including morphology (AFM) of the final device and control measurements before and after deposition, are included in the manuscript. Most importantly, repeatability of their observation should be addressed and, in the ideal case, confirmed.

We appreciate the reviewer's suggestion. As requested, we have prepared three more 1L samples with chirality of (9,4) and (8,6), and performed measurements before and after transfer as shown in Fig. R6. After the transfer of 1L WSe₂, the three samples show several red shifted IX peaks, consistent with the 1L sample in Fig. 1.

Fig. R6. (left) Optical microscope image of the (top) (8,6) CNT and (middle, bottom) two (9,4)/1L WSe₂ heterostructures. The PL spectra (middle) before and (right) after forming the heterostructures. The excitation energy is adjusted to E₂₂ and the power is 10 μW for middle, right middle panels and 5 μW for the right top panel and 0.3 μW for the right bottom panel.

To include these additional experiments, we have added Fig. R6 as Supplementary Fig. 4 and added a sentence on Line 100, Page 5:

“Such low-energy peaks are also observed in other monolayer WSe₂ heterostructures with type-II band alignment (see Supplementary Fig. 4).”

Although there are no published reports of AFM data discussing the sub-nanometer scale (CNT diameter is ~1 nm) morphology on suspended monolayer 2D material to our knowledge, we attempted to evaluate the morphology of two of the samples in Fig. R6. However, the intrinsic ultrahigh softness of the suspended monolayer WSe₂ flake posed significant challenges during the measurement.

For the (8,6) CNT/1L WSe₂ sample which has a wide trench width of 1.5 μm, the WSe₂ flake was severely damaged by the cantilever during the measurement even though we paid attention to using the tapping mode with optimized gain parameters. For the #1 (9,4) CNT/1L WSe₂ sample which has a narrow trench width of 0.5 μm, the AFM data is shown in Fig. R7. The large-area AFM height image (top, left) captures the morphology information, showing the flake dipping into the trench by approximately 4.5 nm. However, the detailed height measurement (top, right) exhibits streaks along with the scanning direction (vertical) within the trench area, which we think does not reflect the intrinsic morphology of the suspended region. Instead, it likely indicates artifacts arising from feedback loop errors and/or adhesive interaction with the cantilever, which are common issues when measuring ultrasoft materials. This interpretation is supported by the amplitude image in Fig. R7 (bottom), which displays a significant variation across the trench area.

We also performed AFM measurements on two more monolayer samples, but similar noise was observed over the trench area. Such an issue prevents us from identifying CNTs beneath the monolayer WSe₂ flakes. Therefore, the detailed morphology information, particularly concerning CNTs, is restricted to samples with thicker WSe₂ layers, which is already included in Supplementary Fig. 17.

Fig. R7. (top) AFM height images of the #1 (9,4) CNT/1L WSe₂ heterostructure. The AFM image for the area in (right) is indicated by the white broken rectangle in (left). (bottom) The AFM amplitude image and line profiles of the (9,4) CNT/1L WSe₂ heterostructure. The line profiles along the red and blue lines in (left) are shown in (right).

- The authors treated the 1L-4L samples on equal footing, but as reviewer 3 pointed out, band structures and carrier dynamics in 1L WSe₂ are quite different from those in thicker films. I strongly feel that the authors should consider performing more careful theoretical calculations to understand the electronic structures of these heterostructures. This could help prevent any misassignment and improve the depth of the manuscript.

We agree that the band structure in 1L WSe₂ is quite different, and thus we have performed additional theoretical considerations as suggested. In our type-II heterostructures, electrons should reside within CNTs while holes should be located in WSe₂. The valence band maxima (VBM) for WSe₂ are therefore of our interest, and their energies at the K and Γ points are extracted from the DFT simulations and plotted in Fig. R8 (left). The energy is higher at the K point for 1L and at the Γ point for thicker ones, resulting in the well-known transition from direct gap to indirect gap. The Γ point shows a substantial shift of 0.515 eV from 1L to 2L, whereas the K point shows a modest increase by 0.036 eV.

Experimentally, making use of additional samples we prepared for this response, we can now compare 1L and 2L WSe₂ for four distinct CNT chiralities (9,4), (8,6), (10,5), (9,8) as shown in Fig. R8 (right). In (9,4) and (8,6) heterostructures, IX peaks are observed for both 1L and 2L with no apparent layer number dependence, indicating a full type-II band alignment. (10,5) heterostructures exhibit a layer number dependence as also shown in Fig. 2d. Only the 2L shows the IX peak, implying a transition from type-II to type-I. In (9,8) heterostructures, IX peaks are absent for both 1L and 2L, consistent with type-I band alignment.

Fig. R8. (left) Calculated VBM energies as a function of WSe₂ layer number. The energies are measured from the vacuum level. (right) The normalized PL spectra for the (9,4), (8,6), (10,5), (9,8) CNT/1L and 2L WSe₂ heterostructures, respectively. The excitation energies are adjusted to E₂₂ energies. A weak emission peak at 1.013 eV in (10,5) CNT/2L WSe₂ sample comes from the E₁₁ emission of an adjacent CNT.

If we assume that the electronic states at the Γ point play an important role, we expect to see band alignment transitions in many chiralities as the VBM at the Γ point shows a large change in energy. For 2L samples, we observe band alignment transition from type-I to type-II as the E_{11} energy is increased by 0.105 eV from (9,8) to (10,5). For (8,6) and (9,4), the relative increase of E_{11} energy from (9,8) is 0.170 and 0.247 eV, respectively, and the valence band energy reduction should be within this energy range. In comparison, the VBM energy at the Γ point decreases by more than 0.5 eV for 1L compared to 2L. This change in VBM is much greater than the change in the E_{11} energy, and therefore we expect the band alignment transitions back to type-I when we replace 2L WSe₂ with 1L WSe₂ in (10,5), (8,6), and (9,4). Experimental results show that the transition is only observed for (10,5), in contradiction with the assumption.

A more reasonable interpretation is that the states at the K point is important for the formation of the interface excitons. The VBM at the K point shows a mere 0.036 eV change from 1L to 2L, and is consistent with the fact that we only observe band alignment transition for (10,5). This interpretation is also supported by the appearance of the IX peaks anticorrelated with exciton transfer. The A exciton in WSe₂, which is formed by electron and hole states at the K point, plays an important role in the exciton transfer as reported in arXiv:2307.07124. In this case, the band alignment with the states at the K point should determine whether exciton transfer is allowed. Because the IX peaks are anticorrelated with exciton transfer, it is implied that formation of interface excitons is also determined by band alignment of the hole states at the K point.

To include these additional discussions, we have added Fig. R8 as Supplementary Fig. 10, which is referred to in Line 154, Page 8.

- With majority of the raw data showing a $g_2 \geq 0.5$, I find the claim of antibunching and single photon emission lack convincing data support. I would recommend the authors providing g_2 data with more apparent single photon emission evidence, or removing this conclusion.

We acknowledge that the value at the dip in the raw data is around 0.5. We have performed an additional photon-correlation measurement on the #2 (9,4) CNT/1L WSe₂ sample in Fig. R6 and obtained a similar value (Fig. R9). The single photon purity is determined by the value of α , and we extract 72%, 89%, and 79% for the samples in Fig. 4b, Supplementary Fig. 15b, and Fig. R9, respectively. Although we feel that this is sufficient experimental evidence to claim single-photon emission, we understand the conservative view by the reviewer. As recommended, we have removed the claim on single-photon emission and refer to quantum emission and antibunching instead.

Fig. R9. (left) A PL spectrum from the (9,4) CNT/1L WSe₂ heterostructure. Peaks at 0.845, 0.958, and 1.100 eV correspond to IX, E_K, and E₁₁, respectively. (right) The second-order correlation statistics for the red IX peak. A longpass filter (0.919 eV) is employed to exclude PL signals from E₁₁. The excitation energy is adjusted to E₂₂ of 1.699 eV and a power of 0.15 μ W. The red line is a fit. From the fitting, we extract τ_A and τ_B as 0.133 and 0.324 ns, and α and β as 0.79 and 0.42, respectively.

- There is a lack of discussion about the origin of these possible IX species at the fundamental level. As mentioned earlier, the potential novelty of this work lies in the origin of these interlayer states. The authors should consider combining their experimental data with detailed theoretical calculations to provide more physical insights, including band structure and electronic structure of the localized IX states.

We thank the reviewer for encouraging us to combine our experimental data and DFT calculations. As suggested, we have provided additional discussion in response to Point 4 above. Together with the discussion on the possible origins of the localization that has been added in the previous round to the second and third paragraphs on Page 12, we believe that we now provide deeper physical insights on interface excitons at a more fundamental level.

- It seems that the current work is a follow up of their previous publication based on the same heterostructure (arXiv:2307.07124). The authors should state clearly how this work distinguishes from their previous publication.

As the reviewer pointed out, both manuscripts study the excitonic physics in mixed-dimensional 1D CNT/2D WSe₂ heterostructures. Specifically, fabrication process of samples and optical characterization methods are common to both.

The two manuscripts are otherwise entirely different. The arXiv:2307.07124 manuscript reports the findings of the exciton transfer process, whereas this manuscript reports the first observation of low-energy interface excitons. The arXiv:2307.07124 manuscript therefore studies samples

with type-I band alignment that exhibit exciton transfer, but this manuscript mainly investigates samples with type-II band alignment required for the appearance of the interface excitons. This can be easily confirmed as the samples mainly discussed in the arXiv:2307.07124 manuscript have different CNT chiralities compared with those in this manuscript.

To clarify the differences of these manuscripts, we have modified the sentence in Line 54, Page 3 as below:

“The chirality dependent bandgap of CNTs can be utilized to tune the band alignment as demonstrated in exciton transfer process [15], allowing for unambiguous identification of excitonic states at the 1D-2D interface.”

Reviewer #3 (Remarks to the Author):

Fang et al. rereview,

The authors provided a strong rebuttal with additional data. They have addressed the significant concerns. I believe the manuscript is sufficiently revised and can be considered for publication. My final request is for the authors to provide $g_2(0)$ data on long-time scales (preferentially plot the time axis on a logarithmic scale) up to micro/mili seconds (similar to Fig.3 in ref.1). This will not only ensure the autocorrelation function is normalized to 1 on long-time delays accurately and provides a better fit for your g_2 values (for instance, plot R6.right might imply there is bunching at -4,+4 seconds and seems like an odd stopping point to normalize the g_2) but also gives very valuable information for theorists to try to model the SPE and the involved level structure.

[1] Patel, Raj N., et al. "Probing the optical dynamics of quantum emitters in hexagonal boron nitride." PRX Quantum 3.3 (2022): 030331.

We sincerely thank the reviewer for stating that we have “provided a strong rebuttal” and that “the manuscript is sufficiently revised”. We are also happy to see that the reviewer can now consider the manuscript for publication in Nature Communications.

In response to the request, we have conducted additional experiments with a new (9,4) CNT/1L WSe₂ sample. The photon correlation results for this sample are presented in Fig. R9, with long-time scale correlations plotted under both linear and logarithmic scales in Fig. R10. It is confirmed that the photon correlation function is accurately normalized to 1 up to dozens of nanoseconds. Notably, the sample exhibits reduced bunching behavior and a rapid antibunching timescale of 0.133 ns, which could be attributed to the use of low laser power. Unfortunately, our current hardware limits us from exploring longer timescales up to micro/milliseconds, as setting sufficient resolution for observing the antibunching dip restricts the time window for the photon correlation.

To indicate that the photon correlation function is normalized to 1, we have added Fig. R10 in

Supplementary Note 16, and we also referred to Ref. [1] given by the reviewer in the same note.

Fig. R10. The long-time second-order correlation statistics for the samples in Fig. R9 under linear and logarithmic scales. The black line for the linear scale and the dots for the logarithmic scale are experimental results. The red lines are fits. Four data points are binned together for the linear scale data and the gray dots.

REVIEWERS' COMMENTS

Reviewer #1 (Remarks to the Author):

The authors have addressed my major concerns. The revision is sufficient. I recommend the publication of the manuscript as it is.